# Self-organization of modular activity in immature cortical networks

Haleigh N. Mulholland [1], Matthias Kaschube [2,3,5] & Gordon B. Smith [1,4,5] ✉

During development, cortical activity is organized into distributed modular patterns that are a precursor of the mature columnar functional architecture. Theoretically, such structured neural activity can emerge dynamically from local synaptic interactions through a recurrent network with effective local excitation with lateral inhibition (LE/LI) connectivity. Utilizing simultaneous widefield calcium imaging and optogenetics in juvenile ferret cortex prior to eye opening, we directly test several critical predictions of an LE/LI mechanism. We show that cortical networks transform uniform stimulations into diverse modular patterns exhibiting a characteristic spatial wavelength. Moreover, patterned optogenetic stimulation matching this wavelength selectively biases evoked activity patterns, while stimulation with varying wavelengths transforms activity towards this characteristic wavelength, revealing a dynamic compromise between input drive and the network's intrinsic tendency to organize activity. Furthermore, the structure of early spontaneous cortical activity – which is reflected in the developing representations of visual orientation – strongly overlaps that of uniform opto-evoked activity, suggesting a common underlying mechanism as a basis for the formation of orderly columnar maps underlying sensory representations in the brain.

A hallmark of the primary visual cortex of primates and carnivores is the columnar, modular organization of neural activity. Here, responses to visual stimuli are both locally correlated among neighboring neurons and globally coordinated in distributed networks that extend over millimeters[1–3], forming a series of active domains that alternate across the cortical surface with a specific wavelength[4]. Such so-called modular organization is clearly evident in the topographic arrangement of selectivity for a variety of visual features, such as stimulus orientation[1–3], ocular dominance[5,6], binocular disparity[7], luminance polarity[8], and direction selectivity[9,10], and is also observed in ongoing spontaneous activity[11,12].

Developmentally, this modular functional organization is also present in spontaneous activity patterns in the immature cortex well before eye-opening and the onset of reliable sensory-evoked responses[11,13,14]. Correlations in modular spontaneous activity at this time appear to serve as a precursor to the developing representations of stimulus orientation[11], suggesting that by coordinating the activity of distant neurons into distributed networks over the course of development, these early modular activity patterns may play a crucial role in establishing and refining the functional networks used for visual perception. Notably, such correlations are present at a time when the long-range axonal connections that eventually link correlated and co-tuned domains[15,16] are still poorly developed[17], and these early correlations do not depend on structured feed-forward inputs[11,13]. This raises the possibility that these early spontaneous activity patterns self-organize during activation of intracortical circuits, such that their

[1]Department of Neuroscience, University of Minnesota, Minneapolis, MN 55455, USA. [2]Frankfurt Institute for Advanced Studies, 60438 Frankfurt am Main, Germany. [3]Department of Computer Science and Mathematics, Goethe University, 60054 Frankfurt am Main, Germany. [4]Optical Imaging and Brain Sciences Medical Discovery Team, University of Minnesota, Minneapolis, MN 55455, USA. [5]These authors jointly supervised this work: Matthias Kaschube, Gordon B. Smith. ✉e-mail: gbsmith@umn.edu

modular structure forms as an emergent property via recurrent interactions and without the need of structured inputs[11]. However, the circuit mechanisms underlying the generation of modular spontaneous activity in the developing cortex remain poorly understood.

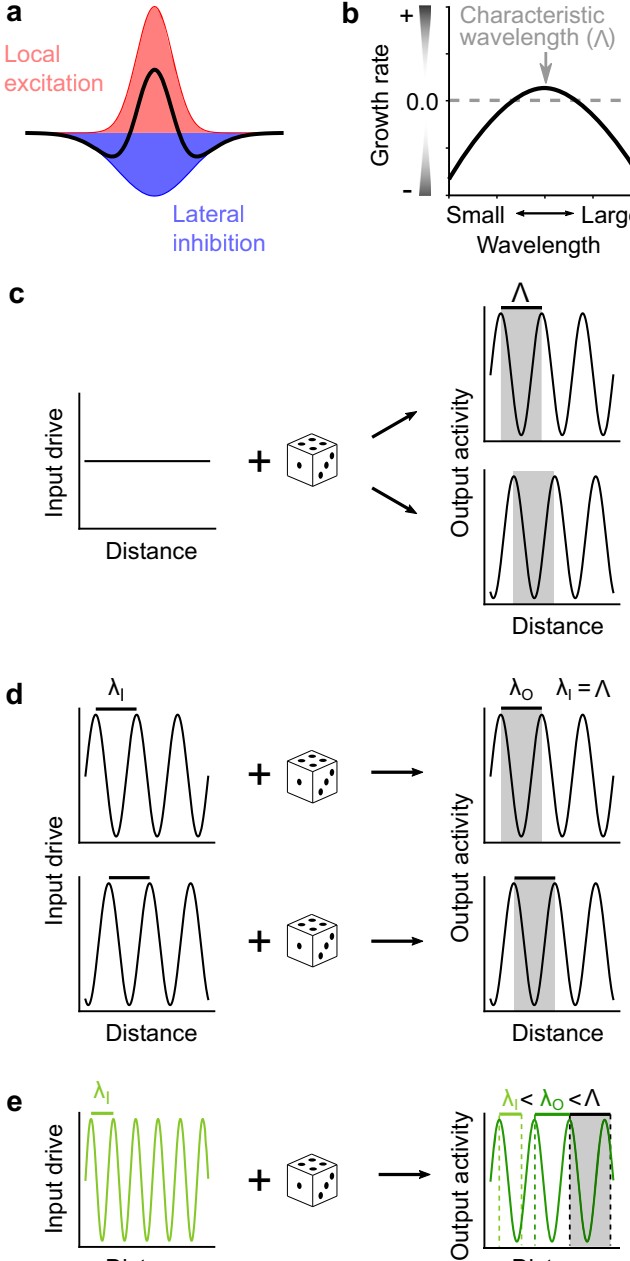

**Fig. 1 | Key predictions of LE/LI mechanism that selectively amplifies a characteristic wavelength of activity patterns. a** The LE/LI mechanism: short-range interactions through a network connected with effective local excitation (red) and lateral inhibition (blue). **b** Networks with this connectivity scheme can selectively amplify activity at a characteristic wavelength, Λ. **c** Simplified one dimensional schematic, showing that uniform input drive produces spatially modular output with regularly spaced peaks and troughs close to the characteristic wavelength Λ of the network (gray bar). The spatial phase of the output patterns is variable, dependent on noise conditions (as indicated by dice). **d** When driven by a structured pattern with an input wavelength ($\lambda_I$) that is consistent with Λ, the output activity is biased towards the input pattern. **e** When driven by a pattern with an input wavelength that is slightly different than Λ (smaller in this example), the resulting wavelength of the output activity ($\lambda_O$) is intermediate, sitting between $\lambda_I$ and Λ.

Numerous computational studies over the past decades have demonstrated that modular patterns can emerge in neural networks with competitive lateral interaction, typically in the form of local excitation and longer-range lateral inhibition (LE/LI, Fig. 1a)[4,18–29]. Included among these are models which recapitulate the modular structure of spontaneous activity in developing cortex[11]. Collectively, this class of models builds upon a theoretical framework first laid out by Alan Turing[30], wherein periodic spatial structures can arise through a dynamical network that shapes patterns with a system-specific, finite characteristic wavelength[31,32]. The mechanism describes at its core the selective, dynamic amplification around this wavelength (Fig. 1b) mediated by a network that combines local self-activation and lateral inhibition[33,34]. This mechanism combines stability with flexibility in that it enables the robust formation of a modular organization – to group functionally related neurons facilitating their signal exchanges – but to achieve this with great flexibility regarding the absolute spatial position of these modules. While originally conceived in the context of morphogenesis during development utilizing diffusible morphogens[30,33,35], iterations upon this basic mechanism have long stood as a theoretical framework for organized activity in the brain. Here, distributed populations of neurons could self-organize their activity to produce modular, large-scale patterns from relatively short-range connectivity and without the need for specific structured inputs. Implementations of this framework have been successfully applied on both developmental timescales—for example to explain the formation of orientation preference maps in visual cortex[4,18,20,23,27–29]—and on the timescale of neural activity—for example to explain the dynamic emergence of modular spontaneous events[11,19,23,24,26], including during early development[11] when horizonal connections are still mostly short-range[17].

The central theoretical feature uniting these models is the hypothesis of functional LE/LI, yet empirical evidence for specific neural connectivity schemes that could support such a mechanism has been scarce (refs. 36–38; but see ref. 39). Although experiments that manipulate the relative strengths of excitation and inhibition on a developmental timescale yield results consistent with LE/LI[40], a direct demonstration of self-organization operating on the timescale of neural activity remains lacking. An alternative explanation holds that specifically organized feed-forward inputs, such as those potentially resulting from interference patterns generated by orderly mosaics of retinal ganglion cells[41–43], would give rise to modular co-activation within the cortex. Other hypotheses hold that instead of emerging spontaneously via selective dynamic amplification at synaptic timescales, modular cortical activity could result from an anatomical scaffold, which has been proposed to specifically link distributed patches of cortex[44]. The specificity of these nascent clustered horizontal connections would then result in the specific co-activation of subsets of modules, yielding both correlated spontaneous activity and the matched tuning properties seen in the orientation preference map[44]. Thus, while having strong explanatory power and serving as the basis for a rich body of computational studies, the role of the LE/LI mechanism in the formation of modular neural activity in the early developing visual cortex has remained unclear, and whether cortical networks enable neural activity to dynamically self-organize at this early developmental timepoint has not yet been tested in vivo.

To determine whether the modular spontaneous activity seen in the immature cortex early in development arises from dynamic intracortical LE/LI interactions, we set out to test three critical predictions of such a mechanism in developing cortical networks by combining simultaneous in vivo widefield calcium imaging with optogenetic stimulation in ferret visual cortex prior to the onset of visual experience. Using both uniform and spatially structured optogenetic stimulation, we confirm these predictions indicating that cortical activity self-organizes into modular patterns that are strongly biased towards a characteristic wavelength. Pharmacological blockade of either

feedforward inputs or cortical synapses supports that the circuit mechanism organizing these patterns resides within cortex itself. Together, these results provide strong evidence that modular activity patterns in the immature cortex arise as an emergent property from local intracortical interactions of the type LE/LI. When we compared these modular patterns to spontaneous activity, we found clear similarities in structure, suggesting that endogenous modular activity also arises via self-organization through a similar LE/LI mechanism.

## Results

### Unstructured optogenetic stimulation of visual cortex evokes structured modular activity

If neural activity self-organizes into modular patterns through a LE/LI mechanism on the timescale of synaptic transmission, this then leads to several critical and testable predictions. Firstly, it predicts a break in the symmetry of neural activity elicited by a spatially uniform input, leading to the transformation of unstructured input into organized and modular outputs that exhibit a characteristic wavelength (denoted by Λ) (Fig. 1b, c). In such a system, a variety of output patterns at this wavelength result from the amplification of any variability in activation that might arise from several sources, including variable initial activation, noisy signal propagation within the network, and additional inputs to the network from other brain regions (collectively summarized by the dice in Fig. 1c–e). Thus, repeated presentations should produce a diverse range of modular patterns from the same uniform input. Secondly, as the LE/LI mechanism specifies a characteristic wavelength Λ for activity but does not restrict the spatial phase of active domains, driving activity with spatially patterned input consistent with Λ should bias the spatial structure of the emerging modular activity to be similar to the input pattern (Fig. 1d). Finally, unlike a rigid scaffold, the LE/LI mechanism flexibly amplifies activity around a characteristic wavelength (Fig. 1b), and thus will transform structured input with a dominant wavelength different from Λ into activity patterns with an intermediate wavelength, shifted towards Λ (Fig. 1e).

We developed a microscope for artifact-free simultaneous widefield calcium imaging and optogenetic stimulation[45]. This microscope allows us to project arbitrary patterns onto the surface of the cortex with high spatial and temporal specificity (Fig. S1), allowing for precise control of large populations of neurons across millimeters of cortex. To simultaneously image and optogenetically stimulate excitatory populations in vivo, we injected layer 2/3 of young ferret visual cortex with 2 viral vectors, one expressing the calcium indicator GCaMP6s[46] and a second expressing a somatically targeted, red-shifted excitatory opsin ChrimsonR[47] (Supplementary Fig. S2). Both viruses expressed under the synapsin promotor, which predominantly labels excitatory cells in juvenile ferrets[48]. Imaging was performed 12–18 days later, between postnatal days 23 and 29 (P23-P29), prior to eye opening (approximately P31) (Fig. 2a).

To test the first prediction, we stimulated the cortex with a spatially uniform full-field optogenetic stimulus and found that it elicited a robust and reliable rise in GCaMP signal in the cortex that emerged rapidly and was time-locked to the stimulation (Fig. 2b, c, Supplementary Movie 1). Strikingly, this neural activity was highly non-uniform, with full-field stimulation evoking modular, patterned activity that extended over several millimeters of cortex (Fig. 2d), with the specific pattern of activity varying from trial-to-trial (Fig. 2e). These opto-evoked modular patterns consisted of regularly spaced active domains and were highly reminiscent of the modular structure seen previously in both spontaneous[11,14] and visual grating-evoked activity[4,49,50].

To quantify the magnitude and the regularity of the spacing of active domains of individual opto-evoked events, we defined a modularity parameter capturing the amplitude of modulation in the autocorrelation function (see Fig. 2f and Methods). We found that this quantity showed a significant increase over baseline (Fig. 2g, opto-evoked: 0.12 (+/−0.01), baseline (200 ms prior to stimulus onset): 0.04 (+/−0.01), $p < 0.01$, $n = 8$ animals, Wilcoxon signed rank (WSR) test), demonstrating that organized patterns with pronounced, regularly spaced modules reliably emerge from spatially uniform inputs. In addition, increasing stimulus power increased both the modularity of the pattern and the amplitude of modules (see Methods), with both showing a sigmoidal relationship as a function of input power, consistent with the presence of a dynamic instability when the input drive passes a threshold level of neural activation (Supplementary Fig. S3a–e).

We estimated the spatial wavelength from the autocorrelation of each individual activity pattern (Fig. 2f) and found that across animals the wavelength was constrained to a narrow band (Fig. 2h, mean wavelength: 0.82 mm +/−0.01; mean standard deviation of wavelength: 0.15 mm +/−0.03, $n = 8$ animals) and was invariant to stimulus intensity (Supplementary Fig. S3g, r = 0.003, $p = 0.985$). Such a narrow distribution of event wavelength is consistent with the presence of a characteristic wavelength, as expected with activity produced in a network through a LE/LI mechanism. Note that the wavelength of opto-evoked events was highly similar to that of spontaneous events that we recorded in the same animal (Supplementary Fig. S4), showing that this characteristic wavelength also applies to activity patterns that naturally occur at this stage in development.

To examine the apparent variety of activity patterns elicited by the same uniform stimulus input, we next computed trial-to-trial correlations across opto-evoked events and sorted these correlations by hierarchical clustering, revealing considerable trial-to-trial variability (Fig. 2i). Further, we quantified the number of linear dimensions which explain the majority of this variance and found that opto-evoked activity resides in a moderately sized dimensional space (Fig. 2j; dimensionality = 10.9 (+/−0.1), event number matched ($n = 40$ events), $n = 8$ animals, mean (+/− SEM)). The diversity and variability of opto-evoked patterns across trials is consistent with trial-to-trial differences in noise amplified through a LE/LI mechanism (Supplementary Fig. S5a–c). Notably, this variability was not reflective of an inability of the developing cortex to respond reliably to a given input, as change of luminance visual stimuli through the closed eyelid evoked consistent patterns of activity that revealed modular ON/OFF preference functional maps (Supplementary Fig. S7), consistent with those seen previously in older animals[8]. Altogether, these results demonstrate that early cortical networks are capable of transforming unstructured inputs into a rich repertoire of modular activity patterns.

### Optogenetic stimulation with structured patterns selectively biases cortical activity

In self-organizing activity generated through a LE/LI mechanism, small random fluctuations in activity are amplified around a characteristic wavelength Λ to produce modular activity. Critically, if cortical networks operate in this manner, then selectively modulating the input drive by imposing a spatially structured optogenetic stimulation at the characteristic wavelength Λ onto the cortex should produce cortical activity patterns that reflect the structure of the stimulation pattern, as we illustrate in a recurrent network model implementing the LE/LI mechanism (Supplementary Fig. S5d–f, see Methods).

To test this in vivo, we generated optogenetic stimuli consisting of random spatial patterns (Fig. 3a) approximating the spatial wavelength of spontaneous activity. Patterned stimuli produced robust, modular responses that showed partial similarity with the stimulus on individual trials (Fig. 3b) and were strongly overlapping in trial-averaged activity (Fig. 3c, Supplementary Fig. S6). Trial-to-trial correlations showed that responses to a given stimulus pattern were selective (Fig. 3d, Supplementary Fig. S8a;), and structured opto-stimulation drove more consistent response patterns than uniform stimulation (Supplementary Fig. S8b). Importantly, both individual opto-evoked responses (Fig. 3e, f) and trial-averaged responses (Fig. 3g) showed

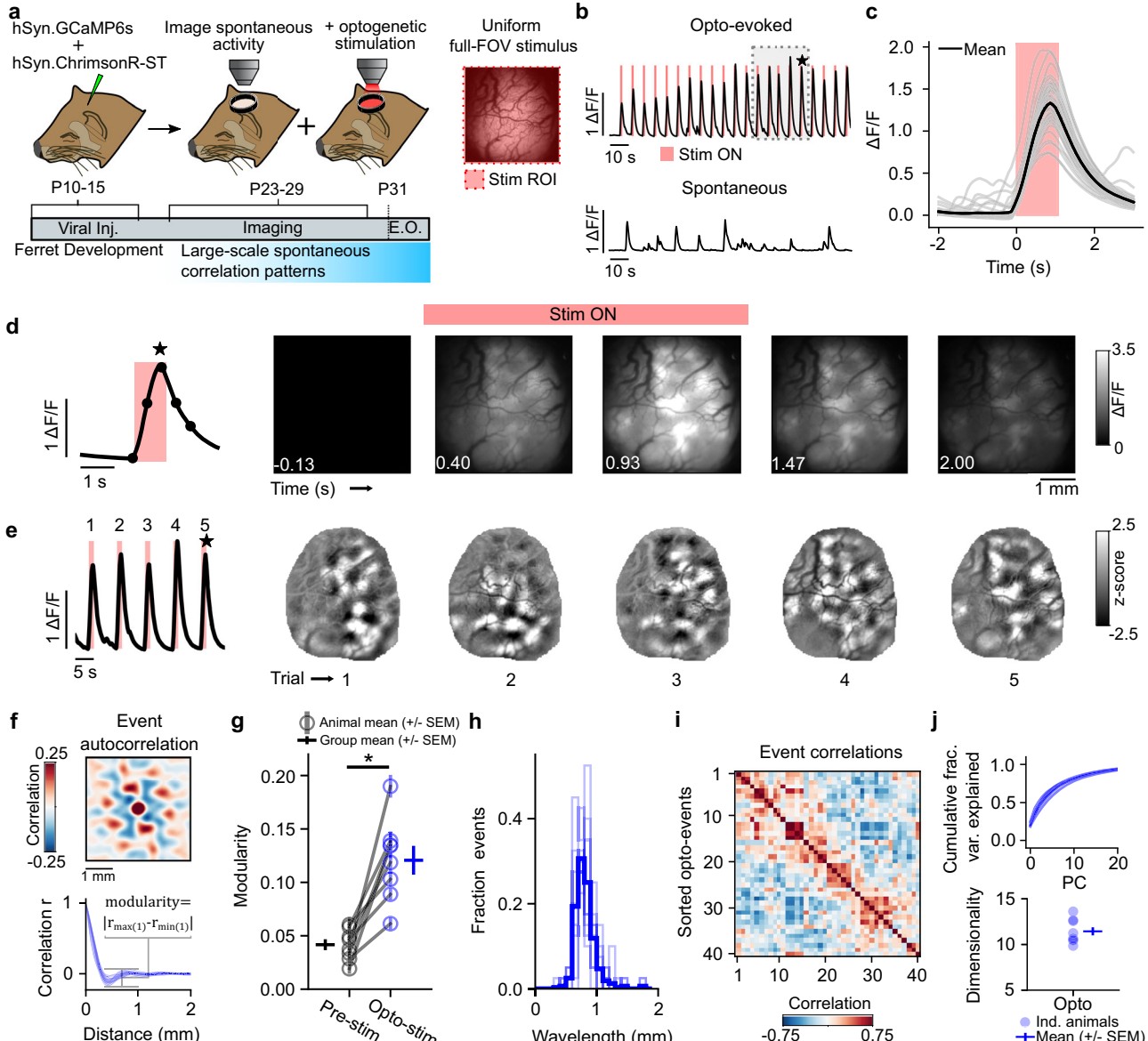

**Fig. 2 | Uniform optogenetic stimulation evokes diverse patterns of modular activity. a** Experimental schematic. Simultaneous optogenetic stimulation and calcium imaging in developing ferret kits, prior to eye opening (E.O.) **b**, **c** Full-field optogenetic stimulation reliably evokes calcium responses. **b** Mean activity across FOV for opto-evoked (top) and spontaneous (bottom) experiments within the same animal. Gray box indicates region shown in **e**. **c** Stimulus-triggered average response. **d** Time series of a single trial example demonstrating that uniform optogenetic stimulation evokes modular activity. **e** Opto-evoked activity is reliably modular and shows variability in pattern across trials. (*Left*) Mean calcium activity across FOV of 5 sequential trials. (*Right*) Spatially filtered events of corresponding trials. Star indicates trial shown in **d**. **f** Opto-evoked events shows regularly spaced modules in both 2D autocorrelation (top, single event) and 1D radial average of

individual events (bottom, $n = 40$ events from one example animal). **g** Opto-evoked activity shows significant modularity compared to pre-stimulus baseline in all animals (mean +/−SEM across trials ($n = 40$) for each animal ($n = 8$ animals). * indicates significant $p$ value, $p = 0.008$, Wilcoxon sign rank test (WSR)). **h** Distribution of wavelengths for all events for all animals $n = 40$ trials per animal, 8 animals. Thin lines show individual animals, thick line is mean across animals. **i** Sorted opto-evoked events spatial correlations from a single example animal. **j** Cumulative variance explained of principal components (top) and dimensionality (bottom) for all animals show that opto-evoked activity resides in a moderately sized dimensional subspace of all possible activity patterns (dimensionality = 10.9, mean +/− SEM, $n = 8$ animals).

significant similarity to the spatial structure of the input pattern applied to the cortex (Fig. 3f, Individual trial similarity: Stimulus pattern vs evoked response = 0.133 (+/−0.024), Trial shuffled = 0.030 (+/−0.011), $p = 0.016$, WSR, $n = 7$ animals; Fig. 3g, Trial averaged similarity: Stimulus pattern vs evoked response = 0.423 (+/−0.038), Trial shuffled = −0.038 (+/−0.023), $p = 0.016$, WSR, $n = 7$ animals). The greater similarity seen in trial-averaged responses compared to individual trials was consistent with model results in a regime with strong noise within the cortical network (Supplementary Fig. S5d−h). Thus, our results show that the specific pattern of network activity can be

influenced by the structure of input to the network when it matches its characteristic wavelength, consistent with the ability of developing cortical circuits to self-organize activity by amplifying biases in the inputs into the network.

## Cortical networks transform structured inputs towards a characteristic wavelength

Networks driven by a LE/LI mechanism produce activity by amplifying most strongly activity modes in a band around the characteristic wavelength Λ (Fig. 1b). As the LE/LI mechanism operates in a regime

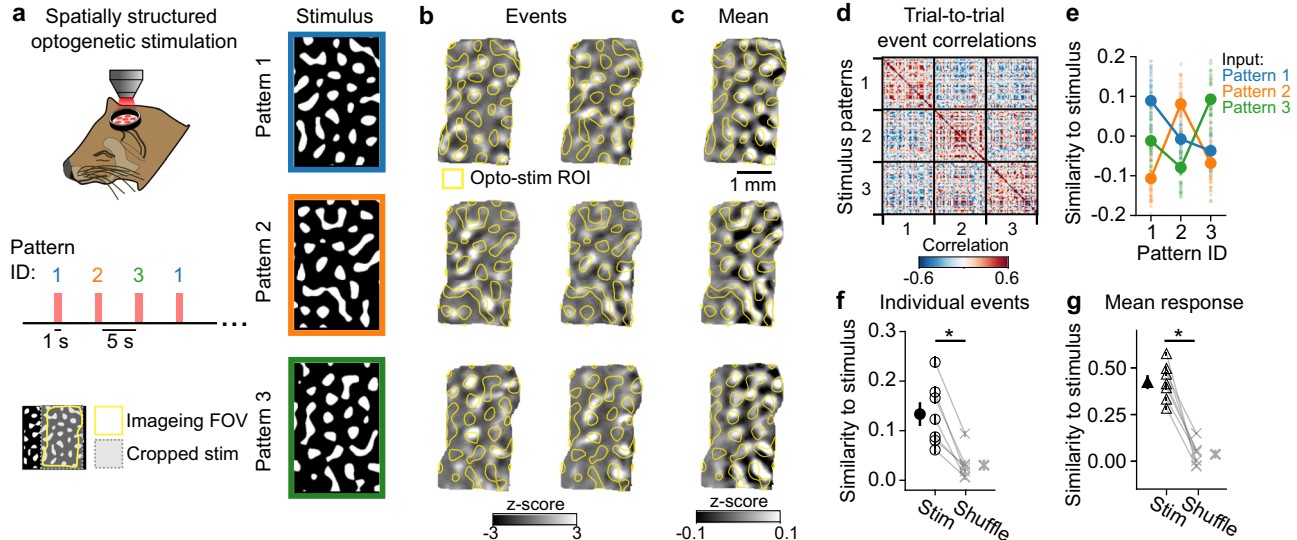

**Fig. 3 | Structured optogenetic stimulation selectively biases cortical responses. a** Experimental schematic. Patterned optogenetic stimulation with 3 arbitrary patterns that approximate the wavelength of endogenous activity. **b** Individual opto-evoked events are variable and show partial overlap with stimulus input. **c** Mean response across trials shows strong overlap with stimulus pattern. **d** Trial-to-trial event correlations show elevated similarity for responses evoked by the same stimulus pattern. **e** Opto-evoked events specifically reflect the input stimulus pattern. Data from same example animal in **d**. Across animals, both opto-evoked individual event responses (**f**) and trial averaged responses (**g**) were significantly more similar to their respective stimulus input pattern than trial shuffled controls. * indicates *p* values < 0.05. **f** Each open circle shows the mean (+/−SEM) similarity to input pattern across individual trials pooled across stimulus patterns for each animal (*n* = 40 trials per *n* = 3 stimulus patterns, total 120 trials), compared to trial-shuffled controls. Closed circle indicates across animal mean (+/−SEM), *n* = 7 animals, *p* = 0.008, one-sided WSR. (**g**) Open triangle shows similarity of mean response pattern to input averaged (+/−SEM) across *n* = 3 stimuli. Closed triangle indicates across animal mean (+/−SEM), *n* = 7 animals, *p* = 0.008, one-sided WSR.

where networks are neither weakly recurrent (where output patterns are largely specified by input) nor rigidly scaffolded (where the set of possible outputs is fixed by the network), inputs with wavelengths that differ from Λ should produce outputs that lie at an intermediate wavelength, between Λ and the input. For example, in the case of limited noise, driving these networks with an input pattern that has a dominant wavelength slightly smaller (or larger) than this characteristic wavelength can more strongly activate modes closer to this smaller (or larger) wavelength, resulting in output activity reaching a compromise between the input pattern and the network's inherent tendency to self-organize activity at a characteristic wavelength. However, our results above (Fig. 3, Supplementary Fig. S5) suggest the presence of a LE/LI mechanism operating in a network with considerable noise, so we therefore sought to understand how changes in the input wavelength affect activity patterns in such a regime. To this end, we studied our recurrent network model in a high noise regime (as in Supplementary Fig. S5e, f). We found that the transformation from input to output wavelength is strongly evident in individual trial responses, which show a wavelength highly similar to the network's characteristic wavelength regardless of input (Fig. 4a, b). Averaging across model responses, instead, revealed the biasing influence of input wavelength, with activity showing the expected compromise between the stimulus input and the network's preferred wavelength (Fig. 4a, c).

To test whether in vivo cortical networks similarly transform the wavelength of input activity patterns towards its characteristic wavelength, we optogenetically stimulated visual cortex with random bandpass patterns of varying wavelength (Fig. 4d). We found that from trial-to-trial, these stimuli evoked a diverse series of patterns (Fig. 4e), where, regardless of the wavelength of the input stimulus, the wavelength of the resulting cortical activity on individual trials was highly similar to Λ (Fig. 4g,i, *p* > 0.3 for all wavelength bins, stimulus patterns pooled across animals, Wilcoxon rank sum test). However, when we calculated the mean response across trials of a given wavelength, we saw that the wavelength of this average response reflected an intermediate value, falling between the spacing of the input pattern and Λ (Fig. 4h, i), as predicted by the model (compare Fig. 4c). These results demonstrate that patterns of cortical activity do not merely reflect the structure of cortical inputs, nor are they fixed by cortical networks on a rigid scaffold. Instead, early cortical networks dynamically transform structured inputs into activity patterns shifted towards the characteristic wavelength, while at the same time reflecting the biasing influence of the input wavelength, thus confirming a critical test of the LE/LI mechanism in vivo.

## Modular activity can arise through intracortical interactions

The ability to elicit modular activity patterns consistent with a self-organizing system by directly activating V1 neurons optogenetically suggests that the circuit mechanisms that organize this activity also reside within local intracortical circuits in V1. To rule out the alternative possibility that opto-evoked modular structure requires instructive feedforward inputs into the network, we silenced the lateral geniculate nucleus (LGN), the primary input to visual cortex, with the GABA(A) agonist muscimol (Fig. 5a). After confirming that visually evoked responses were eliminated in the cortex (Supplementary Fig. S9), we repeated optogenetic stimulation of V1, finding that uniform stimulation still evoked strong responses that remained highly modular (Fig. 5b–e; modularity: baseline = 0.10 (+/−0.01), muscimol = 0.09 (+/−0.01), *n* = 40 events, *p* = 0.138, Wilcoxon rank-sum test, mean (+/−SEM)). These results indicate that cortical circuits are sufficient to dynamically sculpt modular patterns from a uniform input.

In order to directly assess whether activity propagating through intracortical circuits is required for opto-evoked activity to become modular, we next blocked excitatory synaptic activity in our imaging field through bath application of the glutamatergic antagonist kynurenic acid (KYN) to the cortex (Fig. 5f). Uniform optogenetic stimulation reliably drove an increase in calcium activity, presumably through direct activation of cells expressing ChrimsonR and GCaMP, yet this activity remained spatially uniform and failed to become modular, exhibiting a flat spatial structure that was markedly different

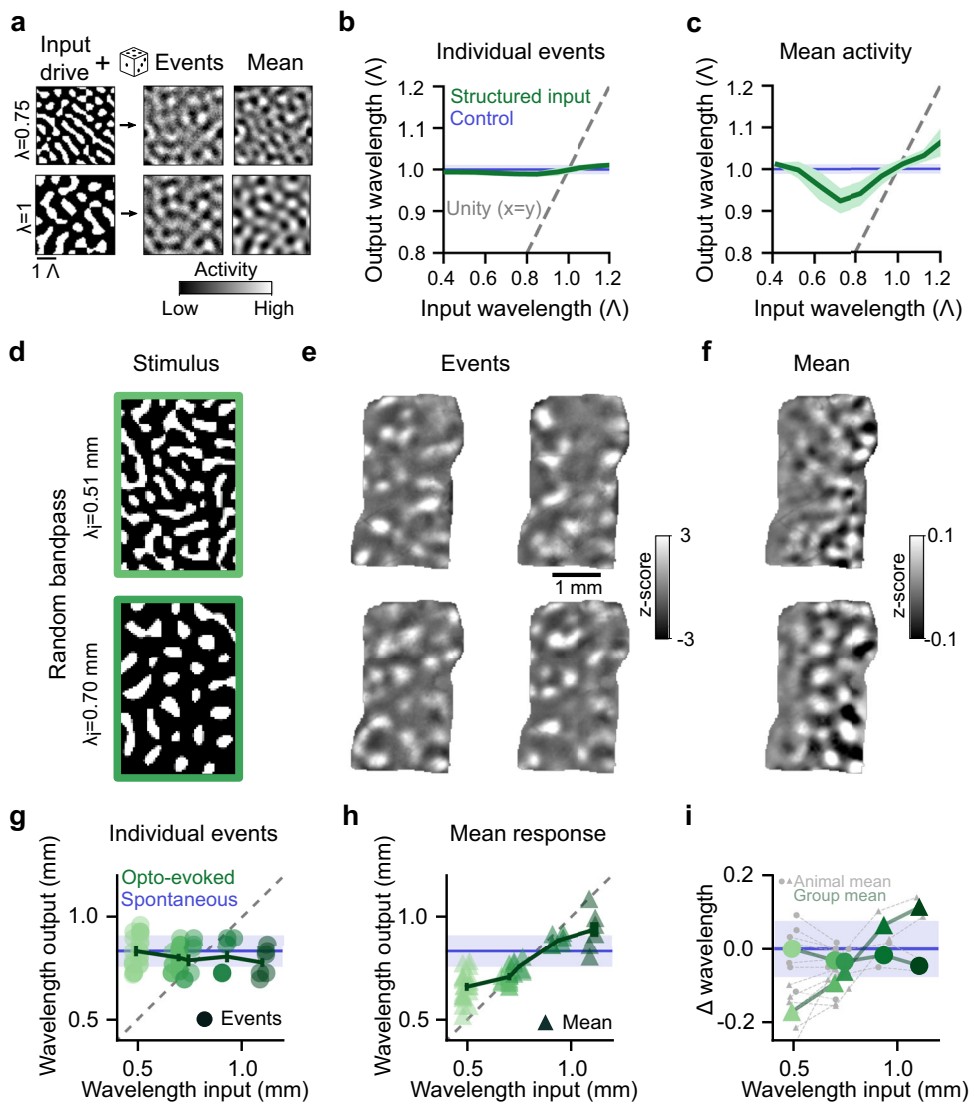

**Fig. 4 | Cortical networks shift activity towards the characteristic wavelength.** **a–c** Computational model shows transformation of input wavelength predicted by a LE/LI mechanism in the presence of strong noise. **a** Structured input drive into network generates modular outputs that show wavelength transformations. **b** Wavelength of individual events for structured input drive of varying wavelength (green, mean (+/−SD)) closely matches the characteristic wavelength (Λ) generated by white-noise input (blue, mean (+/−SD)), largely independent of the dominant wavelength of structured input. Data plotted in units of Λ. **c** Mean activity across model events (green, mean (+/−SD)) shows a wavelength in between the input wavelength (gray dashed line) and Λ (blue, mean (+/−SD)). **d** In vivo optogenetic stimulation with random patterns of varying wavelength. **e** Individual events show opto-evoked, modular activity with wavelength that appears unaffected by the input wavelength. **f** Mean activity across events shows wavelengths shifted towards input wavelength. **g** Across a range of input wavelengths, individual opto-evoked events show a wavelength that closely matches the characteristic wavelength observed in spontaneous activity (Λ, blue line = mean (+/−SD)). Dots show individual animals, mean (+/−SEM) across trials, n = 90 stimulus patterns from 8 animals. **h** Same as **g**, but for the wavelength of mean response. Mean activity is biased away from input wavelength (gray line) towards Λ (blue). **i** Change from Λ (blue, mean (+/−SD)) for individual events (circles) and mean activity (triangles). Gray = individual animals, green = mean across animals.

from opto-evoked activity in control conditions (Fig. 5g–j; baseline modularity: 0.19 (+/−0.01), KYN modularity: 0.01 (+/−0.001), n = 40 events, p < 0.001, Wilcoxon rank-sum test, mean (+/−SEM)). Together, these results show that developing cortical networks require intra-cortical synaptic transmission but do not require specific structure in feedforward inputs in order to produce modular patterns, further demonstrating that modular activity in the cortex self-organizes, emerging out of dynamic network interactions that reside within cortical circuits.

## Uniform opto-stimulation evokes patterns similar to endogenous cortical activity

Our results demonstrate that the developing cortex can self-organize to produce modular activity patterns with a characteristic wavelength. Several lines of evidence suggest that similar self-organizing mechanisms govern the modular structure of sponta-neous activity in the early cortex: 1) computational models imple-menting LE/LI interactions recapitulate the modular structure of spontaneous activity[11]; 2) like opto-evoked activity (Fig. 5), modular spontaneous activity is independent of feedforward inputs[11,13]; 3) spontaneous events show a narrow distribution of wavelengths and closely match both the wavelength and modularity of activity evoked by uniform optogenetic stimulation (Supplementary Fig. S4). We therefore sought to compare the modular spatial pat-terns of spontaneous activity and uniform opto-evoked events, in order to assess whether a similar network mechanism for self-organizing activity could underlie spontaneous activity in the early cortex.

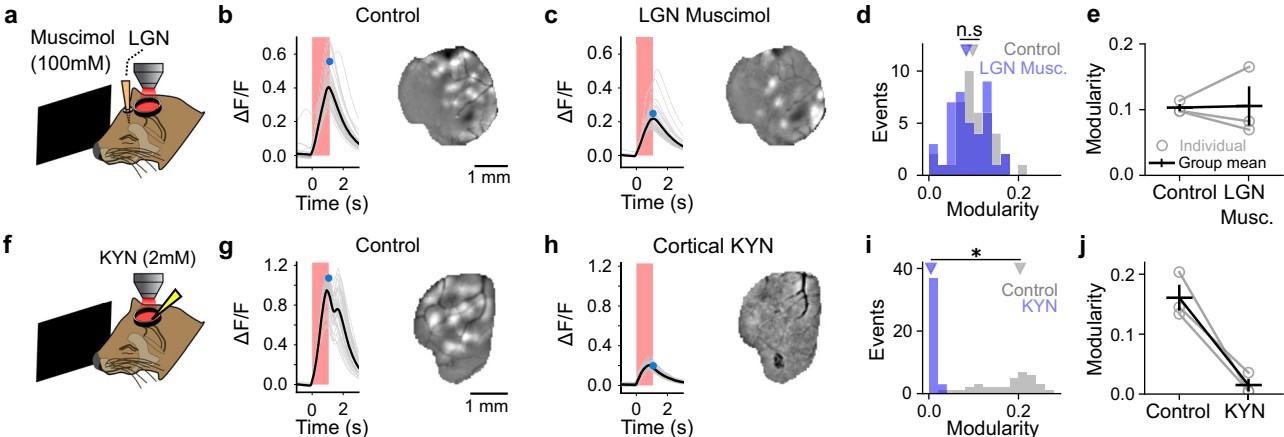

**Fig. 5 | Intracortical circuitry is sufficient and necessary for the emergence of opto-evoked modular patterns. a**–**e** Opto-evoked activity remains modular after silencing feedforward activity from the LGN. **b** Stimulus triggered average response across FOV, (*left*) and modular evoked response (at time indicated with blue dot) (*right*) prior to silencing LGN. **c** Same as **b**, after LGN silencing. **d**, **e** LGN silencing does not alter modularity. **e** Distribution of event modularity for experiment shown in **b** and **c** (*n* = 40 events per condition, *p* = 0.138, Wilcoxon rank-sum test, triangles indicate median). **e** Summary across 3 experiments, (gray = individual animals, black = mean (+/−SEM)). **f**–**j** Blocking intracortical glutamatergic synaptic activity eliminates modular opto-evoked activity. **g** Control condition prior to cortical KYN application. **h** Opto-evoked activity in presence of KYN shows increase in calcium response, but activity fails to become modular. **i**, **j** Cortical KYN significantly reduces opto-evoked event modularity. **i** Modularity distribution for experiment shown in **g** and **h** (*n* = 40 events per condition, *p* < 0.001, Wilcoxon rank-sum test). **j** Summary across 3 experiments (gray = individual animals, black = mean (+/−SEM)).

To address this, we first directly compared the spatial structure of spontaneous and uniform opto-evoked events by calculating the spatial correlation between all pairs of activity patterns. We frequently identified pairs of opto-evoked and spontaneous events with highly correlated spatial structure (Fig. 6a), and for the majority of opto-evoked patterns we could identify at least one spontaneous event showing a similar pattern of active domains (Fig. 6b, 89.38% (+/−0.03) of opto-events with correlations >2 standard deviations over surrogate events (see Methods), *n* = 8 animals, mean (+/−SEM)). Across all trials, the strength of these correlations was significantly greater than chance (Fig. 6c; *p* < 0.001 vs surrogate for 8 of 8 animals, Kolmogorov–Smirnov test) indicating a strong degree of overlap between opto-evoked patterns and spontaneous activity. Similarly, the millimeter-scale correlations across events that are a hallmark of spontaneous activity in visual cortex[11,13,14] are likewise evident in opto-evoked activity, where they also show significant similarity to spontaneous correlations (Supplementary Fig. S10), providing further evidence for a common network mechanism.

We next asked whether the activity patterns evoked by uniform optogenetic stimulation span the full repertoire of spontaneous activity, or rather reflect a biased subset of activity patterns. When opto-evoked events were projected onto spontaneous principal components (PCs), we found no significant difference in the weights of individual PCs across opto-evoked and cross-validated spontaneous activity (Fig. 6d, *p* > 0.05 for all components that explain 75% of total variance). Across animals, the PCs that explained the greatest amount of the spontaneous variance were highly correlated with the components that explained the greatest amount of opto-evoked variance (Fig. 6e, spontaneous vs opto-evoked PCs: $R^2$ = 0.830, spontaneous vs surrogate PCs: $R^2$ = 0.102, all PCs pooled across all animals). When compared to the activity patterns evoked by change in luminance visual stimuli, we found these patterns occupied a distinct activity space, which poorly overlapped with both opto-evoked and spontaneous activity (Supplementary Fig. S7). Importantly, this indicates that the modular patterns evoked by uniform stimulation are not the result of direct activation of the retina by our stimulation light. Together, these results show that spontaneous and opto-evoked activity reside in highly similar activity subspaces. This strongly indicates that both uniform stimulation and spontaneous activity in the cortex self-organize utilizing the same circuit mechanisms to produce similar modular activity patterns.

Prior work has shown that spontaneous activity is low dimensional, indicating that it is constrained to a subset of all possible activity patterns[11,14]. Although such low dimensionality is consistent with LE/LI mechanisms that incorporate a degree of heterogeneity in the system[11], it also raises the question of whether activity patterns in the developing cortex are so constrained that only patterns lying within the spontaneous activity space can be produced. In fact, such a constraint would not be expected from cortical activity with self-organization dominated by LE/LI mechanisms, which instead predicts that the patterns evoked by a structured input near the characteristic wavelength can deviate somewhat from those seen in spontaneous activity (or during uniform stimulation) by reflecting also the structure of the input drive. The possibility of such deviation is consistent with our observation that luminance evoked visual responses poorly explain uniform opto-evoked responses (Supplementary Fig. S7) and also with the biases we observe for structed stimulation in Figs. 3 and 4. In order to test this possibility explicitly, we examined the relationship of the activity patterns evoked through structured optogenetic stimulation at the characteristic wavelength (Fig. 3) with spontaneous patterns, to determine if these opto-evoked patterns lie entirely within spontaneous activity, or rather evoke novel patterns of activity consistent with a dynamic network based on a LE/LI mechanism.

To accomplish this, we first found for each individual, spatially structured opto-evoked event (with wavelengths approximating Λ, Fig. 3) its maximally correlated corresponding spontaneous event, which we refer to as its "best match" (see Methods). The similarity of this best-match event reflects the maximum similarity a given opto-evoked event has with the repertoire of spontaneous events. We observed that for all structured opto-evoked events for a given stimulus, we could find a best-match spontaneous event with positive correlations and some overlapping features. Interestingly, we found that these evoked events also exhibited novel features that did not align with the best match spontaneous pattern (Fig. 6f, Supplementary Fig. S11a–c). To test to see if these patterns could be explained by endogenous activity alone, we again projected opto-evoked patterns onto spontaneous principal components. This time we found significant deviations in the weights of projections onto spontaneous PCs (Fig. 6g), and that across all stimuli presented, spontaneous PCs explained a significantly lower total fraction of variance for opto-evoked events compared to spontaneous controls (Fig. 6h, total

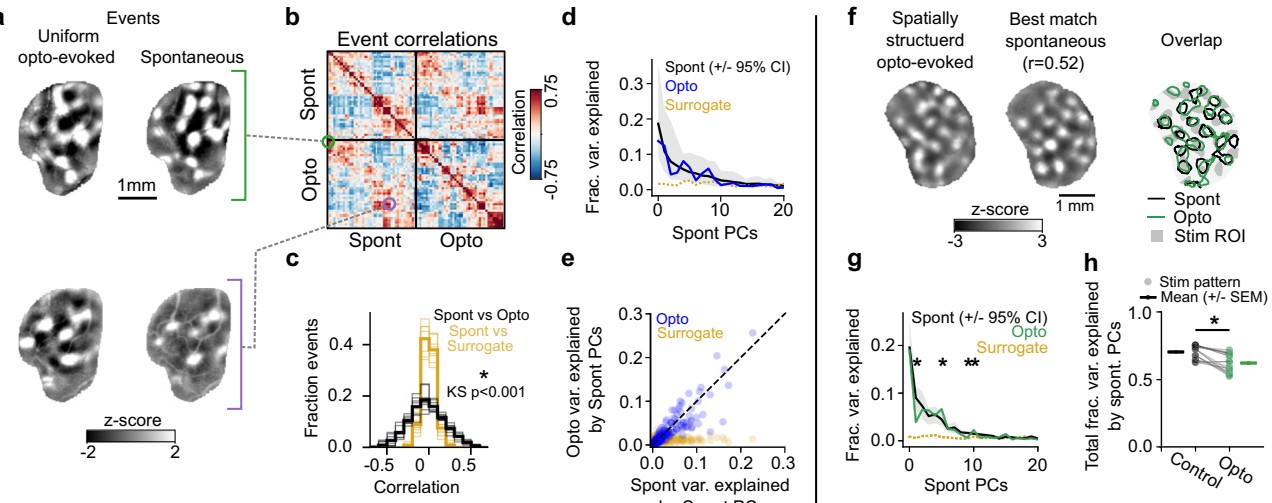

**Fig. 6 | Uniform stimulation evokes events that overlap with spontaneous activity, while spatially structured stimulation evokes events that reveal novel features. a** Example pairs of spontaneous events and uniform full-field opto-evoked responses with highly similar spatial patterns of activity. **b** Correlation matrix of spontaneous and opto-evoked events from a single example animal, sorted into clusters of similar patterns. Note the presence of strong positive correlations between many spontaneous and opto-evoked events. **c** Correlation values for spontaneous vs opto-evoked events are significantly stronger than correlations between spontaneous and surrogate patterns (thin, transparent lines = individual animals; dark, opaque lines = mean across group.) $p < 0.001$ vs surrogate for 8 of 8 animals, Kolmogorov–Smirnov test. **d** Projection of uniform opto-evoked patterns onto spontaneous principal components (blue) explains similar variance as for cross-validated spontaneous events (black, mean +/− 95% CI), while surrogate events (yellow) are poorly explained by spontaneous principal components. Events from single example animal shown in **a**. **e** Spontaneous principal components which explain the majority of the variance of spontaneous activity also explain the

majority of variance in opto-evoked activity (blue). The variance of surrogate activity explained by spontaneous PCs is not correlated with the variance explained of spontaneous events (yellow). **f** Individual trial spatially structured opto-evoked event (*Left*) and its maximally correlated spontaneous event (*Middle*). (*Right*) Overlay of contours of active domains for spontaneous (black) and opto-evoked (green) events and the opto-stimulus pattern (gray). **g** Projection of opto-evoked patterns (driven by stimulus in **f**) onto spontaneous principal components, showing significant deviations in fraction of variance explained compared to spontaneous control (green, spatially-structured opto-evoked explained variance. Black, mean cross-validated ($n = 100$ simulations) spontaneous explained variance (+/−95% confidence intervals). * indicates opto-evoked components that fell outside the 95% confidence intervals, see Methods). **h** Total fraction of the variance explained by spontaneous PCs for spatially structured opto-evoked datasets and cross-validated spontaneous controls ($n = 21$ stimuli, pooled across animals. Total variance summed across first 40 PCs. $p < 0.001$, WSR test).

fraction variance explained: opto = 0.62 (+/−0.02), spontaneous cross-validated control = 0.70 (+/−0.01), $n = 21$ stimulus patterns, pooled across animals; $p < 0.001$, WSR). This unexplained fraction of variance is consistent with model predictions (Supplementary Fig. S11d, e). Together, this indicates that while opto-evoked patterns share some overlap with endogenous activity, structured stimuli can evoke patterns with novel features, suggesting a dynamic compromise between the structure of cortical inputs and the patterns the cortex endogenously produces. Furthermore, given that the structure of the cortical input is capable of both biasing outputs and introducing new structures, the most parsimonious explanation for the similarity in structure and dimensions between uniform opto-evoked and spontaneous activity is that both are generated through similar intracortical mechanisms acting upon similarly unstructured input drive to the cortex.

## Discussion

Network interactions of the form LE/LI are a robust and flexible means of achieving modular organization within a system, transforming poorly structured inputs into complex, well-structured, and large-scale output patterns. By direct in vivo optogenetic stimulation and pharmacological manipulations, we show that cortical networks transform both uniform and patterned inputs with various wavelengths into spatially distributed and regularly spaced modular activity patterns with a characteristic wavelength, and that this cortical transform takes place in a specific manner that confirms critical predictions of a LE/LI mechanism. Thus, our results argue that cortical activity patterns are not rigidly scaffolded by a set of genetically or anatomically defined preestablished structures, nor do they require feedforward input in

order to become modular, but rather emerge dynamically through short-range intracortical interactions.

Self-organization based on local facilitation and lateral inhibition was first mathematically described by Turing[30]. While not the only mechanism for spontaneous pattern formation, it appears to be ubiquitous in nature[33,51,52] in both animate and inanimate systems across vastly different scales (e.g. refs. [53–56]). Our results are consistent with such a mechanism operating at neural activity timescales to shape the structure of spontaneous activity via the selective dynamic amplification of patterns near a system specific characteristic wavelength. Such selective amplification requires the cortical network to operate not far from the critical point of pattern formation (see Methods), but does not require fine tuning close to this critical point. For simplicity, we used a model in a subcritical regime to study selective amplification in the presence of structured input drive and noise, but currently we cannot rule out the possibility that the immature cortex instead operates in a supercritical regime. In fact, our observation of a nonlinear increase in the amplitude of a modular pattern as a function of stimulus intensity (Supplementary Fig. S3) could possibly hint towards a supercritical regime and the presence of a linear instability underlying the pattern amplification process. Regardless of whether above or below the critical point, a LE/LI mechanism would provide the developing cortex with a robust yet flexible way to generate modular functional organization in cortical networks.

The considerable overlap between uniform opto-evoked responses and spontaneous activity—together with their similar dependence on intracortical activity (Fig. 5 and refs. [11,13])—suggests that the circuit mechanisms that are being engaged by uniform opto-stimulation of the cortex are likely to be the same ones that produce spontaneous

activity in the early developing cortex. Given that correlations in early spontaneous activity predict features of the future organization of orientation preference domains[11], our results suggest that these same mechanisms also underlie the structure and modular organization of functional maps in visual cortex. Therefore, by providing strong evidence for the presence of a LE/LI mechanism in the developing cortex, this work justifies a critical assumption underlying a long history of theoretical models that seek to explain the formation of modular functional maps in V1 of carnivores and primates (e.g. refs. 4,11,18,20–29).

Notably, the ability of inputs to flexibly bias activity patterns is a key feature of LE/LI mechanisms, which permit cortical networks to produce highly organized sets of patterns while also allowing specific inputs to influence the structure of output activity. Indeed, we show that optogenetically stimulating with spatially structured stimuli can bias the spatial structure of evoked patterns and elicit patterns that deviate from those seen in spontaneous activity. Our results show that developing cortical networks can selectively, but at the same time flexibly, respond to the structure of their inputs. In this way the repeated activation of a given set of inputs over the course of development could drive Hebbian plasticity, strengthening these inputs as well as connections between modular activated regions across the cortex. Our findings thus suggest how developing cortical networks may be able to build the reliable responses seen in the mature cortex, where specific visual stimuli elicit specific and selective modular responses[2,3,5–10]. This highlights a key developmental benefit of self-organization via a LE/LI mechanism: with only short-range lateral interactions, the cortex can robustly and yet flexibly achieve large-scale organization that can serve as a template for the orderly maps of sensory representations.

Collectively, our results strongly support the presence of a LE/LI mechanism operating in the cortex, implying the existence of a circuit correlate of this mechanism. However, the exact neural circuitry responsible remains unclear. At the time in development that we examined (approximately 1 week prior to eye-opening), the scale of horizontal connections in the cortex is on the order of several hundred microns[17,38,39], which suggests that some combination of excitatory and inhibitory horizontal connections could account for the characteristic wavelength we observe. A simple implementation of a LE/LI interaction is by direct short-range excitatory and long-range inhibitory connections. While there is some work supporting this circuit connectivity in developing ferrets[39], other studies find that for the majority of synaptic connections in visual cortex, the spatial extent of excitation tends to be longer or roughly equal to inhibition[36–38], thus arguing against a purely anatomical circuit implementation. However, models have shown that in certain parameter regimes modular patterns can arise by generating a functional LE/LI motif with effective lateral suppression even if the anatomical range of excitation exceeds that of inhibition, for instance through fast polysynaptic excitatory-to-inhibitory signaling[27,57,58]. Thus, while the specific circuit implementation remains elusive, our results indicate the presence of an effective local facilitation and lateral suppression structure in cortical networks. Our observation that the spatial wavelength is virtually unchanged as a function of laser power (Supplementary Fig. S3g) may provide a useful constraint in the search of candidate circuit implementations[59]. Future studies may be able to uncover the precise mechanistic implementation by analyzing the connectomics of V1 or using optogenetic tools to selectively manipulate local circuit elements within the cortex.

Similar to the attractor models that have been proposed previously to explain spontaneous activity in V1 (exemplified by Goldberg et al.[24]), the dynamics of pattern formation in the model used in this study predict interactions over relatively long (on the order of hundred milliseconds) time scales[60]. Such long time scales have been suggested to be unrealistic in the mature cortex[61], where balanced amplification[62] could be a plausible alternative. However, this may not be the case in the immature cortex, where inputs are temporally less precise[63]. Although our calcium imaging lacks the temporal resolution to definitively address the question of timescales in attractor models, future experiments utilizing voltage imaging or multi-electrode arrays could provide key insights, and may also help determine whether the effective intracortical interactions change during cortical maturation, for instance transitioning from a more recurrent to a more input-dominated regime.

Why some animal species, such as primates and carnivores, exhibit modular functional organization of neural activity, while others, such as rodents and lagomorphs, do not and instead exhibit 'salt-and-pepper' organization[64], is a longstanding mystery in systems neuroscience[65,66]. In animals with large brains, modularity has been proposed to be an efficient way to build circuits by minimizing wiring length[67–69] (although see ref. 70), which may have not been evolutionarily advantageous in species with 'salt-and-pepper' organization. However, brain size is not always a good predictor of functional organization, and a clear rule has remained elusive[65,66]. Additionally, new evidence from visual[71,72] and optogenetic[73] stimulation in mouse V1 indicates that rodents my exhibit some degree of modular functional organization after all, but on a scale of 50–200 microns. This opens the interesting possibility that rodent cortical circuits may be utilizing a similar LE/LI mechanism but on a smaller scale, and future research will have to investigate whether this is the indeed the case and, if so, how this mechanism is implemented in the rodent brain.

Altogether, our work provides direct evidence that immature cortical networks self-organize neural activity, testing in vivo long held predictions stemming from computational modeling. Our findings support that large-scale patterns of modular activity emerge from intracortical interactions, which arise through a dynamic network of local facilitation and lateral suppression. Notably, the mechanisms that lead to the emergence of these patterns need not be specific to the visual cortex and could serve as a more universal mechanism for cortical organization throughout the brain, including the entorhinal[74,75] and prefrontal[76,77] cortices (see ref. 61 for review). Indeed, recent work has shown that other brain regions— including primary auditory and somatosensory areas, as well as association areas such as posterior-parietal and prefrontal cortex —in immature ferret cortex also exhibit modular activity patterns with quantitative similarity to V1, suggesting a common underlying circuit mechanism during initial development[78]. Future work will have to assess whether the LE/LI mechanism we demonstrate in visual cortex generalizes to these other cortical areas.

## Methods
### In vivo experiments
**Animals.** All experimental procedures were approved by the University of Minnesota Institutional Animal Care and Use Committee and were performed in accordance with guidelines from the US National Institutes of Health. We obtained 10 male and female ferret kits from Marshall Farms and housed them with jills on a 16 h light/8 h dark cycle. No statistical methods were used to predetermine sample sizes, but our sample sizes are similar to those reported in previous publications.

**Viral injection.** Viral injections were performed as previously described[79] and were consistent with prior work[11,14]. Briefly, we microinjected a 1:1 ratio of AAV1.hSyn.GCaMP6s.WPRE.SV40 (Addgene) and the somatically targeted AAV1.hSyn.ChrimsonR.mRuby2.ST (University of Minnesota Viral Vector and Cloning Core) into layer 2/3 of the primary visual cortex at P10–15, approximately 10–15 days before imaging experiments. This approach resulted in widespread labeling with both GCaMP and Chrimson. At the cellular level, the populations of GCaMP and Chrimson labeled cells were largely overlapping (Supplementary Fig. S2), although the relative strength of expression of both GCaMP and Chrimson varied within cells, as would be expected from stochastic effects resulting from dual

labeling with two AAV viruses. For injection surgery, anesthesia was induced with isoflurane (3.5–4%) and maintained with isoflurane (1–1.5%). Buprenorphine (0.01 mg/kg) and glycopyrrolate (0.01 mg/kg) were administered, as well as 1:1 lidocaine/bupivacaine at the site of incision. Animal temperature was maintained at approximately 37 °C with a water pump heat therapy pad (Adroit Medical HTP-1500, Parkland Scientific). Animals were also mechanically ventilated and both heart rate and end-tidal CO2 were monitored throughout the surgery. Using aseptic surgical technique, skin and muscle overlying visual cortex were retracted. To maximize area of ChrimsonR expression, two small burr holes placed 1.5–2 mm apart were made with a hand-held drill (Fordom Electric Co.). Approximately 1 μl of virus contained in a pulled-glass pipette was pressure injected into the cortex at two depths (~200 μm and 400 μm below the surface) at each of the craniotomy sites over 20 min using a Nanoject-III (World Precision Instruments). The craniotomies were filled with 2% agarose and sealed with a thin sterile plastic film to prevent dural adhesion, before suturing the muscle and skin.

**Cranial window surgery.** On the day of experimental imaging, ferrets were anesthetized with 3–4% isoflurane. Atropine (0.2 mg/kg) was injected subcutaneously. Animals were placed on a feedback-controlled heating pad to maintain an internal temperature of 37–38 °C. Animals were intubated and ventilated, and isoflurane was delivered between 1% and 2% throughout the surgical procedure to maintain a surgical plane of anesthesia. An intraparietal catheter was placed to deliver fluids. EKG, end-tidal CO2, and internal temperature were continuously monitored during the procedure and subsequent imaging session. The scalp was retracted and a custom titanium headplate adhered to the skull using C&B Metabond (Parkell). A 6–7 mm craniotomy was performed at the viral injection site and the dura retracted to reveal the cortex. One 4 mm cover glass (round, #1.5 thickness, Electron Microscopy Sciences) was adhered to the bottom of a custom titanium insert and placed onto the brain to gently compress the underlying cortex and dampen biological motion during imaging. The cranial window was hermetically sealed using a stainless-steel retaining ring (5/16-in. internal retaining ring, McMaster-Carr). Upon completion of the surgical procedure, isoflurane was gradually reduced (0.6–0.9%) and then vecuronium bromide (0.4 mg/kg/h) mixed in an LRS 5% Dextrose solution was delivered IP to reduce motion and prevent spontaneous respiration.

**Simultaneous optogenetics and calcium imaging mesoscope.** To achieve simultaneous widefield calcium imaging and targeted optogenetic stimulation, we built upon previous designs[80] to construct a custom-built mesoscope (Supplementary Fig. S1, see ref. [45]). Epifluorescent calcium imaging was illuminated using a 470 nm LED (Thorlabs M470L5), which was reflected with a long-pass 495 nm dichroic mirror (Chroma T495lpxr, 50 mm) and focused onto the imaging plane using a Nikon objective (B&H Nikon AF NIKKOR 50 mm f/1.4D lens). Emitted light was focused onto the imaging sCMOS camera (Prime BSI Express, Teledyne) using an additional Nikon tube lens (B&H Nikon 105 mm f/2 D-AF DC lens) and an achromat lens (Thorlabs, ACN254-040, f = 40 mm). GCaMP emission was collected using a GFP bandpass filter (Semrock, 525/39).

To maximize the range of laser stimulus power onto the surface of the cortex and minimize the spectral bandwidth of the stimulation light in order to mitigate light artifacts, optogenetic stimulation was driven by a 590 nm continuous wave laser (Coherent MX 590 nm STM; CW). Laser power intensity was modulated using an acousto-optic modulator (Quanta-Tech MTS110-A3-VIS; AOM), which controlled the temporal aspects of the optogenetic stimulation (stimulus onset, duration, frequency, intensity, and waveform). The maximum power density at the imaging plane ranged from 14 to 16 mW/mm². The first-order diffracted beam from the AOM was coupled to a 400 μm multi-

mode fiber (Changchun New Industries (CNI)), passed through a speckle remover (CNI), and into a DMD pattern illuminator (Mightex Polygon1000), which shaped the spatial aspects of the optogenetic stimulus. This patterned light was expanded using tube lens (f = 100 mm) and reflected to the imaging plane with a short-pass 567 nm dichroic mirror (Thorlabs DMSP567L, 50 mm). A photodiode was mounted behind the dichroic to provide precise tracking of stimulus onset times. Light artifacts from optogenetic stimulation were prevented from reaching the camera with a notch filter (594/23, Thorlabs NF594-23) in the collection pathway. Focal distances were adjusted for both the imaging and the stimulation pathways to be parfocal.

**Widefield epifluorescence and optogenetic stimulation.** All imaging of spontaneous activity was done in young animals (P23–29) prior to eye-opening (typically P31 to P35 in ferrets). Widefield epifluorescence imaging was performed with μManager (version 2.0.0-gamma1)[81]. Images were acquired at 15 Hz with 2 × 2 on camera binning and additional offline 2 × 2 binning to yield 512 × 512 pixels. Prior to optogenetic stimulation, baseline spontaneous activity was captured in 10-min imaging sessions, with the animal sitting in a darkened room facing an LCD monitor displaying a black screen.

Optogenetic stimulation was similarly delivered in the absence of visual stimulation, and the animal's eyes were shielded from the stimulation laser to prevent indirect stimulation of the retina. Analysis of the opto-evoked events further confirmed this, as we found that visually-evoked events poorly explained the spatial structure of opto-evoked events (Supplementary Fig. S7). For all experiments opto-stimulation was delivered at 10 mW/mm², except those that investigated the effect of varying the power of the stimulus intensity (Supplementary Fig. S3). Opto-stimuli were presented for 1 s duration with a 5 s interstimulus interval. For uniform full-field optogenetic stimulation, the whole FOV of the DMD illuminator was activated. For patterned stimulation (see below), black and white bitmap images to be projected onto the surface of the cortex were made using Polyscan software (version 1.2.2, Mightex) or Matlab (Mathworks). To map the region of the cortex that was responsive to optogenetic stimulation, we stimulated the cortex with a 4 × 4 grid of approximately 1×1 mm squares moving sequentially over the FOV. We frequently saw a robust response to optogenetic stimulation across the imaging FOV (Supplementary Fig. S12), and animals that failed to show significant responses to optogenetic stimulation in an area >1 mm² were excluded from this study.

**Spatially structured optogenetic stimuli.** We generated artificial structured patterns to project onto the surface of the cortex by bandpass filtering white noise (size 256 ×256 pixels) at varying wavelengths. Bandpass filtering was applied in the frequency domain by a hard cutoff outside the band defined by $f_{low}$ and $f_{high}$. For simplicity, below we provide these cutoffs in units of pixels in the frequency domain, from which frequencies can be obtained through (f-1)l/256, where l is the resolution of pixel per mm in real space (l = 65 pixels/mm in this case). We binarized these patterns by setting a threshold at the 68th percentile pixel values, producing isolated blobs with a specific wavelength. To test the specificity of opto-evoked activity, for each wavelength we generated three artificial patterns, and then alternated in stimulating with each pattern (3 patterns with 40 trials each, 1 s duration, 5 s interstimulus interval). All tested animals were stimulated with patterns aimed to be smaller than the characteristic wavelength of the network ($f_{low}$ = 40 pixels, $f_{high}$ = 100 pixels; wavelength of approximately 0.5 mm, given) and patterns that approximated the characteristic wavelength of the network ($f_{low}$ = 60 pixels, $f_{high}$ = 100 pixels; wavelength of approximately 0.7 mm). For a subset of experiments, we used narrow bandpass patterns to sample a larger range of wavelengths (5 wavelengths varying $f_{low}$ from 40 to 120 pixels with step

size = 20 and setting $f_{high} = f_{low} + 4$, wavelength approx. 0.48 to 1.10 mm, $n = 2$ animals).

**Visual stimulation.** Visual stimuli were delivered on an LCD screen placed approximately 22 cm in front of the eyes. All animal eyelids were closed, except for animals used for LGN silencing experiments ($n = 3$ animals) whose eyelids were manually opened prior to imaging. Full-field change-in-luminance stimuli were used to evoke ON and OFF responses, with a Michelson contrast of 1. Stimuli were presented using Psychopy software(2020.2.8)[82] for 5 s ON, 5 s OFF. Individual ON/OFF evoked-events were calculated by averaging evoked responses over the first 2 s following stimulus onset, and ON/OFF maps were calculated by taking the average ON or OFF response across trials, with difference maps showing the difference between ON − OFF responses.

**LGN silencing.** To deliver the GABA(A) agonist muscimol to the LGN, we drilled a craniotomy window (4–11 mm from the midline, 7–11 mm from lambda) prior to imaging. Dura was retracted and exposed cortex was covered and sealed with 2% agarose. To locate and map the LGN, a 5 MΩ electrode (FHC) was driven approximately 7 mm down perpendicularly into the brain using a micromanipulator. An alternating ON/OFF full-field visual stimulus was presented, and successful location of the LGN was identified by multi-unit or single-unit spike responses to change-in-luminance stimulation. The electrode was retracted, and replaced with a pulled glass pipet filled with 100 mM muscimol. Five pulses of 50 nL each were pressure injected into the LGN at 3 depths along the LGN vertical axis (spaced approximately 500 um apart) using a glass-pulled pipet tip. Effective silencing was confirmed by measuring the loss of ON and OFF responses in the visual cortex (Supplementary Fig. S9).

**Kynurenic acid experiments.** To silence propagating synaptic activity within our imaging field of view, we removed the coverslip cannula from the imaging window and bath applied approximately 100 mL of 2 mM kynurenic acid (KYN) to the surface of the cortex. After waiting approximately 5 min for the drug to take effect, we then delivered full-field optogenetic stimulation to the cortex, using the same stimulus parameters as above.

**Histology and confocal imaging.** For a subset of animals, following imaging animals were euthanized and transcardially perfused with 0.9% heparinized saline and 4% paraformaldehyde. The brains were extracted, post-fixed overnight in 4% paraformaldehyde, and stored in 0.1 M phosphate buffer solution. Brains were cut using a vibratome in 50 μm coronal sections, which were then imaged on a confocal microscope (Nikon AX R).

**Data analysis**

**Signal extraction for widefield epifluorescence imaging.** Image series were motion corrected using rigid alignment and a region of interest (ROI) was manually drawn around the cortical region of GCaMP expression, excluding major blood vessels. The baseline fluorescence ($F_0$) for each pixel was obtained by applying a rank-order filter to the raw fluorescence trace with a rank 70 samples and a time window of 30 s (451 samples). The rank and time window were chosen such that the baseline faithfully followed the slow trend of the fluorescence activity. The baseline-corrected spontaneous activity was calculated as:

$$\Delta F/F_0 = \frac{(F - F_0)}{F_0} \tag{1}$$

**Event detection and preprocessing.** Spontaneous: Detection of spontaneously active events was performed as previously described[11,14]. Briefly, we first determined active pixels on each frame using a pixelwise threshold set to 5 s.d. above each pixel's mean value across time. Active pixels not part of a contiguous active region of at least 0.01 mm² were considered 'inactive' for the purpose of event detection. Active frames were taken as frames with a spatially extended pattern of activity (>80% of pixels were active). Consecutive active frames were combined into a single event starting with the first high-activity frame and then either ending with the last high-activity frame or, if present, an activity frame defining a local minimum in the fluorescence activity. To assess the spatial pattern of an event, we extracted the maximally active frame for each event, defined as the frame with the highest activity averaged across the ROI.

Opto-evoked: Opto-evoked evoked events were detected by taking the frame at opto-stimulus offset. Modular activity was reliably time-locked to the stimulus, and we made no effort to search for peaks to minimize the potential of our results being contaminated by ongoing spontaneous activity.

To preprocess data, all events were mean activity subtracted and filtered with a Gaussian spatial band-pass filter ($\sigma_{low} = 26$ μm and $\sigma_{high} = 195$ μm). The mean activity of opto-evoked events was non-modular (Supplementary Fig. S12a), and mean subtraction helped normalize differences in baseline fluorescence between spontaneous and opto-evoked events.

**Modularity and estimation of event wavelength Λ.** To estimate the wavelength and modularity of individual calcium events, we calculated the spatial autocorrelation of the Gaussian highpass filtered image (Fig. 2f, *top*). We then took the radial average of the autocorrelation to get a 1-dimensional autocorrelation function (Fig. 2f, *bottom*). The wavelength $\Lambda$ of the event was calculated as twice the distance to the first minimum from the origin. Modularity is a measure of the regularity of the spatial arrangement of the pattern. It was calculated by finding the absolute difference in correlation amplitude between the first minimum and the subsequent maximum of the 1-dimensional autocorrelation.

**Modular amplitude.** In order to estimate the amplitude of modular peaks, we measured the average spectral power within a band centered around the characteristic wavelength, defined as the average wavelength of spontaneous activity within each animal. We controlled for FOV size across animals and minimized orientation effects by cropping each unfiltered opto-evoked event to a 2 mm diameter circular mask centered within the imaging ROI, and then reduced the DC component of the image by subtracting the mean ΔF/F across the cropped event frame. From this, we computed the Fourier transform of each event, took the squared modulus and the radial average to get the 1D power spectrum. $F_1$ is the average power within a spatial frequency band centered on the characteristic frequency, i.e. the interval $[2\pi/(\Lambda - 0.06 \text{ mm}), 2\pi/(\Lambda + 0.06 \text{ mm})]$.

**Trial-to-trial variability of opto-evoked events.** To determine the trial-to-trial variability in opto-evoked activity, we computed the event-wise Pearson's correlation between calcium events (Fig. 2i). To visualize clusters of events with similar structure, we performed hierarchical clustering of the correlation matrix (linkage threshold set as half the maximum pairwise distance). Hierarchical clustering was calculated in Python using SciPy's library (version 1.7.3) of hierarchical clustering functions.

When computing the event correlation matrices for spatially structured opto-evoked activity (Fig. 3d), we did not cluster these events but instead organized the matrix by stimulus pattern ID. The trial-to-trial correlation (Supplementary Fig. S8) was summarized as the mean correlation across all pairs of trials driven by the same stimulus pattern (within pattern), compared to pairs of trials not driven by that specific pattern (across pattern). Controls were estimated by random trial shuffling of the stimulus IDs and taking the average

correlation of event number matched trial pairs, then finding the mean across 100 random shuffle simulations.

**Similarity of opto-evoked activity and optogenetic stimulus pattern.** To quantify how similar opto-evoked activity was to its specific stimulus input pattern, we calculated the spatial correlation between each opto-evoked event with each stimulus pattern. The similarity between the i-th event ($A_i$) and the j-th stimulus pattern ($S_j$) is:

$$\rho_{i,j} = \text{corr}(A_i, S_j) \tag{2}$$

where corr is the Pearson's correlation over space. Thus, for each event we calculated how similar it was to the specific pattern it was driven with and can compare this to stimulus patterns that it was not driven with (Fig. 3e). For quantifying how similar the mean trial response was to the stimulus input, the same approach was used, except that $A$ corresponds to the trial averaged pattern. For comparing across animals (Fig. 3f, g) we took the mean across the three stimulus conditions when stimulus ID matched event ID. We estimated a control by trial shuffling the event IDs, to determine how much a given evoked response overlapped with the stimulus pattern by chance (average across 100 random shuffle simulations).

**Similarity between opto-evoked and spontaneous events.** To determine the similarity between individual opto-evoked events and baseline spontaneous calcium events, we computed the event-wise Pearson's correlation between all events from both datasets. To visualize clusters of events with similar structure (Fig. 6b), we performed hierarchical clustering of the correlation matrix and then segregated the opto-evoked and spontaneous events and sorted by these cluster labels. Since the total number of spontaneous events typically outnumbered the opto-evoked events, for presentation purposes we randomly subsampled an opto-event number matched group of spontaneous events and showed their correlations with opto-evoked activity. Statistical analysis of event correlation distributions included the full correlation matrix of all spontaneous and opto-evoked events.

To estimate the amount of spatial correlation expected by chance, we generated a set of surrogate events by randomly flipping, rotating, and shifting opto-evoked events, thereby maintaining the statistics of the images but disrupting any consistent spatial relationships. Rotation angle drawn from a uniform distribution between 0° and 360° with a step size of 10°, translated shifts were drawn from a uniform distribution between ±450 μm in increments of 26 μm, independently for x and y directions. Reflection occurred with a probability of 0.5, independently at the x and y axes at the center of the ROI. We then computed the surrogate events vs spontaneous event correlation matrix and compared the distributions of correlations for opto-evoked and surrogate control data (Fig. 6c). We used the Kolmogorov–Smirnov test to quantify the similarity of distributions. Individual opto-evoked patterns were considered highly similar to spontaneous events if their correlation was greater than 2 standard deviations over the mean surrogate correlation.

To determine whether the amount of overlap between spatially structured opto-evoked activity significantly deviated from the amount of overlap that could be expected to occur naturally in spontaneous activity, we found each opto-evoked event's best matching spontaneous pattern. To control for finite sampling size, best matching pairs were found by finding the maximum event-wise Pearson's correlation between each opto-evoked event and a randomly subsampled, event number matched subset of spontaneous events ($n = 40$). To estimate the null distribution, we did a permutation test finding the average maximum correlation between this same subset of spontaneous events and a separate, equally sized randomly

subsampled subset of spontaneous events ($n$ simulations = 500), which was used to find the 95% confidence interval.

**Pixelwise correlation patterns.** Correlation patters were calculated using either spontaneous events or opto-evoked events. Correlation patters were calculated as previously described[11,14]. Briefly, we down sampled each spatially filtered event to 128 × 128 pixels. The resulting events were used to compute the correlation patterns as the pairwise Pearson's correlation between all locations $\boldsymbol{x}$ within the ROI and the seed point ($\boldsymbol{s}$)

$$C(\boldsymbol{s}, \boldsymbol{x}) = \frac{1}{N} \sum_{i=1}^{N} \frac{\left(A_i(\boldsymbol{s}) - \langle A(\boldsymbol{s}) \rangle\right)\left(A_i(\boldsymbol{x}) - \langle A(\boldsymbol{x}) \rangle\right)}{\sigma_{\boldsymbol{s}} \sigma_{\boldsymbol{x}}} \tag{3}$$

where $A$ are the events, the brackets $\langle \ \rangle$ denote the average over all events and $\sigma_{\boldsymbol{x}}$ denotes the standard deviation of $A$ over all $N$ events at location $\boldsymbol{x}$.

**Comparison of similarity between correlation networks.** To compare the similarity between spontaneous and opto-evoked correlation patterns within the same animal, we computed the second-order correlation between patterns. For each seed point, we calculated the Pearson's correlation between corresponding correlation patterns, while excluding pixels within a 400-μm radius around the seed point to prevent local correlations from inflating the similarity between the two correlation patterns. To obtain an estimate of the upper bound of similarity within spontaneous activity given a finite sampling size, we randomly split spontaneous events into two groups and separately computed correlations and the second-order correlations between the halves ($n$ simulations = 100). To determine if the observed networks are more similar than chance, we calculated the similarity between the opto-evoked network and a network calculated from surrogate events (as above).

**Principal component analysis and dimensionality of calcium events.** We estimated the linear dimensionality $d_{\text{eff}}$ of the subspace spanned by activity patterns by the participation ratio[83]:

$$d_{\text{eff}} = \frac{\left(\sum_{i=1}^{N} \gamma_i\right)^2}{\sum_{i=1}^{N} \gamma_i^2} \tag{4}$$

where $\gamma_i$ are the eigenvalues of the covariance matrix for the $N$ pixels within the ROI. As the value of the dimensionality is sensitive to differences in detected event number, to estimate the distribution of the dimensionality for each animal, we calculated the dimensionality of randomly sub-sampled events ($n = 40$ events, matched across animals, 100 simulations) and took the median of the distribution.

To determine the amount of variance spontaneous activity can explain of opto-evoked events, we projected opto-evoked variance onto the principal components of spontaneous activity. The variance of a dataset A (opto-evoked) explained by the i-th principal component $\boldsymbol{p}_{i,B}$ of dataset B (spontaneous) is:

$$\text{var}_{i,A} = \frac{\boldsymbol{p}_{i,B}^{T} \cdot \boldsymbol{\Sigma}_A \cdot \boldsymbol{p}_{i,B}}{\text{Tr}(\boldsymbol{\Sigma}_A)} \tag{5}$$

where $\boldsymbol{\Sigma}_A$ is the covariance matrix of dataset A and $\boldsymbol{p}_{i,B}$ is normalized to unit length.

Spontaneous principal component analysis was cross-validated by segregating spontaneous events into two randomly subsampled (without replacement) event matched training groups and test groups. The training group was used to generate the principal component basis set, and the test group was then projected onto the training group components to estimate the corresponding variances $\gamma_i$. To

estimate the null distribution and compute confidence intervals, we performed 100 repetitions of cross-validation. Principal components were computed using the python library Scikit (version 1.0.2)[84].

**Quantification and statistical analysis.** Nonparametric tests were used for statistical testing throughout the study. Random subsampling, bootstrapping, and cross validation was used to determine null distributions when indicated. Center and spread values are reported as mean and SEM, unless otherwise noted. Statistical analyses were performed in Python, and significance was defined as $p < 0.05$. Tests for significance were always two-sided, unless otherwise indicated.

## LE/LI network model

To model our spatially structured optogenetic experiments (Fig. 4, Supplementary Figs. S5, S13), we build upon previous work[11]. We modeled the effects of driving cortical activity with spatially structured input patterns in a rate network with recurrent connections following the scheme of local excitation and lateral inhibition (LE/LI). Pattern formation in such a network typically involves the selective amplification of spatial patterns with a characteristic spatial wavelength $\Lambda$. In a nonlinear system such selective amplification may be caused by a linear instability of a uniform solution and the growth of spatial Fourier modes around the characteristic frequency $k = 2\pi/\Lambda$, sometimes referred to as the supercritical regime (Fig. 1b illustrates this case). However, also in a subcritical regime, below but sufficiently close to this critical point of linear instability and pattern formation, modes around the characteristic frequency are selectively amplified when driven by broad-band input, and these modes decay much slower than those with low or high spatial frequency. For instance, when stimulating such network with spatial white noise, a spatial pattern emerges that is dominated by the characteristic frequency. Since the basic predictions illustrated in Fig. 1 essentially test selective amplification around a characteristic wavelength, they apply to both regimes. For the sake of simplicity, we therefore studied our network model in the subcritical regime, ignoring possible effects due to the saturation of pattern growth (Supplementary Fig. S3). To this end, we used a linear rate network model[85]:

$$\tau \frac{dr(\boldsymbol{x},t)}{dt} = -r(\boldsymbol{x},t) + \mu \sum_{\boldsymbol{y}} M(\boldsymbol{x},\boldsymbol{y}) r(\boldsymbol{y},t) + I(\boldsymbol{x}) \qquad (6)$$

Here, $r(\boldsymbol{x},t)$ is the average firing rate in a local pool of neurons at location $\boldsymbol{x}$, $\tau$ is the neuronal time constant (set to 1), and $M(\boldsymbol{x},\boldsymbol{y})$ is the cortical connectivity from location $\boldsymbol{y}$ to $\boldsymbol{x}$. In the most basic form of the model, the cortical connectivity is defined as a difference of Gaussians

$$M(\boldsymbol{x},\boldsymbol{y}) = \frac{1}{2\pi\sigma_1^2} \exp\left(-\frac{|\boldsymbol{x}-\boldsymbol{y}|^2}{2\sigma_1^2}\right) - \frac{1}{2\pi\sigma_2^2} \exp\left(-\frac{|\boldsymbol{x}-\boldsymbol{y}|^2}{2\sigma_2^2}\right) \qquad (7)$$

where $\sigma_1$ and $\sigma_2$ is controlling the spatial range of the excitatory and inhibitory connections, respectively. The factor $\mu$ is controlling the overall strength of connections, set such that the maximum eigenvalue of the connectivity matrix is equal to 0.99 (hence the system close to the critical point of pattern formation). The characteristic wavelength $\Lambda$ of the network was directly estimated from the peak of the spectrum of $M$ and depends on the connectivity parameters through the expression

$$\Lambda^2 = \frac{\pi^2 \sigma_1^2 (\kappa^2 - 1)}{\ln(\kappa)} \qquad (8)$$

where $\kappa = \sigma_2/\sigma_1$. In all our simulations we used a network size = $60 \times 60$ and set $\sigma_1 = 1.8$ and $\sigma_2 = 3.6$ (in pixels), resulting in a value of $\Lambda = 11.76$ pixels. All modeling results shown in the Figures are expressed in units of $\Lambda$. $I(\boldsymbol{x})$ is the input to cortical location $\boldsymbol{x}$, assumed to be constant in

time (described further below). All solutions were computed at steady-state via direct matrix multiplication.

Previous work[11] has shown that networks with isotropic local excitatory and lateral inhibitory connectivity fail to produce activity with long range correlations as large as observed in vivo, but that adding heterogeneity to the connection weights can produce biologically realistic activity patterns, correlation structure and dimensionality. To introduce heterogeneity in our connectivity, we perturbed our network connectivity by multiplying the connectivity kernel $M(\boldsymbol{x},\boldsymbol{y})$, for any fixed location $\boldsymbol{x}$, with the expression:

$$1 + hG(\boldsymbol{y}) \qquad (9)$$

where $h$ is the strength of the perturbation and $G(\boldsymbol{y})$ is a spatially structured Gaussian random field (computed by bandpass-filtering an uncorrelated Gaussian random field using a difference of Gaussians with $\sigma_{\text{low}} = 2$, $\sigma_{\text{high}} = 6$). Consistent with the network size, $G$ was implemented as a $60 \times 60$ matrix and chosen to be the same for each $\boldsymbol{x}$. Importantly, the effects of network wavelength transformations of stimulus inputs show a similar bias towards the characteristic wavelength of the network (Fig. 4a–c) as would be expected from a network with homogenous, isotropic connectivity ($h = 0$, Supplementary Fig. S13), demonstrating that the predictions of the LE/LI mechanism studied here also hold for networks with more realistic, heterogeneous connectivity ($h = 0.4$).

To simulate the effects of input into the network, to model the in vivo experiments with optogenetic stimulation, the input $I$ consisted of a noise component to reflect the impact of various sources of noise in vivo (represented as a dice in Figs. 1, 4, Supplementary Figs. S5 and S13) and a stimulus component:

$$I(\boldsymbol{x}) = \eta N(\boldsymbol{x}) + \omega S(\boldsymbol{x}) \qquad (10)$$

The noise $N(\boldsymbol{x})$ was sampled from an uncorrelated Gaussian random field (centered at zero), with its amplitude controlled by the coefficient $\eta$. The stimulus or input drive $S(\boldsymbol{x})$ was set to 1 for uniform stimulation (Supplementary Figs. S5a–c and S13b). Structured input drive (Fig. 4a–c, Supplementary Figs. S5d–h and S13c–h) was generated analogously to the stimulus patterns used in vivo by applying, in the frequency domain, a narrow bandpass to spatial white noise, and then binarizing the resulting pattern at the 68-percentile threshold. To model the effect of varying wavelengths, we generated structured inputs with incrementally increasing wavelengths (29 bandpass bins, varying $f_{\text{low}}$ from 1 to 8 with step size = 0.25 and setting $f_{\text{high}} = f_{\text{low}} + 2$, $n = 10$ distinct patterns per wavelength). The amplitude of the structured input is defined by $\omega$. To estimate the effect on wavelength for individual activity patterns and stimulus mean responses (Fig. 4b, c), for each bandpass stimulus pattern we ran 40 simulations varying the input noise while keeping $S$ constant. We then computed the wavelength of the individual output activity patterns and the wavelength of the mean output activity across simulations. The spatial wavelength of the band-pass used is expressed relative to that computed from Eq. (8). To determine the impact of stimulus versus noise amplitude (Supplementary Fig. S5h), $\omega$ was varied from 0.05 to 0.6.

## Reporting summary

Further information on research design is available in the Nature Portfolio Reporting Summary linked to this article.

## Data availability

All source data for figures are provided with this paper. Due to size limitations, all original raw data (imaging files) are available from the corresponding author upon request. Source data are provided with this paper.

## Code availability

Custom code[86] used to run linear rate model computational simulations are available at https://github.com/SmithNeuroLab/uniform_opto. https://doi.org/10.5281/zenodo.10892426.

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

## Acknowledgements

The authors wish to thank Nic Glewwe, Matt Paruzynski, Jack Kapler, and Sophie Bowman for histology and surgical assistance, Deano Farinella and Harishankar Jayakumar for their optical expertise in designing and installing our microscope for imaging and optogenetics, and members of the Smith and Kaschube labs for helpful discussions. All viral vectors used in this study were generated by the University of Minnesota Viral Vector and Cloning Core (Minneapolis, MN). This work was supported by the resources and staff at the University of Minnesota University Imaging Centers (SCR 020997). Funding for this work was provided by National Eye Institute (NIH R01EY030893-01; G.B.S.), National Science Foundation (IIS-2011542; G.B.S.), Whitehall Foundation (2018-05-57; G.B.S.), Bundesministerium für Bildung und Forschung (BMBF 01GQ2002; M.K.), LOEWE Research Cluster CMMS (M.K.), National Institute of Mental Health (MH115688 T32; H.N.M.).

## Author contributions

Conceptualization: H.N.M., M.K., and G.B.S. Methodology: H.N.M., M.K., and G.B.S. Investigation: H.N.M. Formal Analysis: H.N.M. Visualization: H.N.M. Funding acquisition: G.B.S. and M.K. Project administration: G.B.S. Supervision: G.B.S. and M.K. Writing – original draft: H.N.M.; Writing – review & editing: H.N.M., M.K. and G.B.S.

## Competing interests

The authors declare no competing interests.
