## [Peer Review File · Nature Communications]

Self-organization of modular activity in immature cortical networksREVIEWER COMMENTS

Reviewer #1 (Remarks to the Author):

See attachment.

Nature Communications Referee Report for Manuscript 444555

“Self-organization of modular activity in developing cortical networks”

by H. Mulholland et al

September 11, 2023

This paper is not acceptable for publication in its present form but a future version may be acceptable if the manuscript is revised substantially in accord with the comments below.

The goal of this paper is to argue, based on novel data collected and analyzed by the authors, that a broadly applicable theoretical mechanism related to spontaneous pattern formation of physical systems—the so-called Turing instability by which an initially uniform unpatterned medium can become unstable to the formation of a cellular pattern with a characteristic length scale—is key to understanding how a particular kind of cellular neural activity (already well known through the efforts of previous researchers) arises during brain development in a variety of animals that include primates and some other animals like ferrets. That is, the authors use their data to argue that certain similar cellular activity patterns observed in a variety of animal brains can be explained as a neurobiological example of a well-known pattern formation mechanism. If true, this result would connect certain questions and details in brain development to a broad and advanced theoretical and experimental literature of Turing instabilities and pattern formation (developed mainly outside of biology by physicists, chemists, and mathematicians), which would stimulate new insights and new questions to be investigated.

While the data are state-of-the-art, new, and interesting, and the questions of how and why similar cellular activity patterns arise during development in the cortices of different animals are important and interesting, this referee feels the paper fails in several substantial ways to make a convincing argument that a Turing instability is relevant. The following are key weaknesses of the paper that I feel have to be addressed in any revised paper:

1. The paper has serious conceptual flaws. One is that the authors never clearly state what they mean by a Turing mechanism so it is difficult to determine what they are claiming in their paper.

The problem is that Turing’s original calculation in 1953 has been now discussed and somewhat reinterpreted by such a big community of researchers (physicists, mathematicians, chemists, biologists, engineers, and others) that the concept of a “Turing mechanism” has become so broad as to be useless unless authors state exactly what they are talking about, which the authors do not in this paper.

Turing’s original calculation was rather narrow and dealt only with the linear instability (growth of infinitesimal perturbations of all different wavelengths) associated with an infinitely wide continuous medium that had no boundary conditions imposed. (Turing discussed mainly two reacting and diffusing chemicals, although he

did discuss briefly a discrete medium consisting of a periodic ring of cells whose sizes were tiny compared to the wavelength of the instability). Turing showed by his pioneering theoretical calculations that, under certain circumstances (such as varying the ratio of the diffusion constants of two chemicals), a spatially uniform medium could become unstable to exponentially growing spatially sinusoidal modes with a narrow distribution of wavelengths centered on the mode with the fastest growing rate (the so-called critical wavelength of the linear instability).

But a key point is that Turing only discussed the onset of a linear instability and did not discuss (because the mathematics was too hard at the time) what the linearly growing modes would evolve into when the nonlinearities of the evolution equations eventually cause the amplitudes of the growing modes to saturate to have finite values. This saturated nonlinear regime is a new and different problem in pattern formation that Turing did not address and many researchers, including the authors, unfortunately and incorrectly, call the nonlinear saturated pattern a Turing pattern or the result of a Turing mechanism (words used in the current paper).

This is an extremely important point for the authors to clarify in their paper. The authors at no time discuss the linear instability of an initially approximately-uniform patternless piece of cortex and so they never are in a position to observe a classical Turing instability. Instead they are using optogenetics to optically perturb a portion of the visual cortex of a still developing ferret *after* the cortex already has developed an intricate inhomogeneous spatial structure that is not well characterized. (Local excitation and longer-range lateral inhibition indeed exist but this is an extremely crude qualitative observation since the anatomical and physiological details have not been measured of the developing cortex and so are poorly characterized.) That is, the authors are reporting results for a medium (visual cortex) whose spatial symmetry is already broken (is nonuniform). While the results are still interesting, the results lie outside of what Turing has discussed and, in fact, lies outside most pattern formation theory since the nonlinear saturation of unstable modes in a finite disordered inhomogeneous medium or the effects of order-one perturbations of a nonlinear saturated medium is a difficult and only partially understood problem.

So Figure 1 of the paper is greatly misleading. Fig. 1b shows, but fails to explain adequately, the growth rates $\sigma(\lambda)$ for infinitely-wide sinusoidal modes of infinitesimal magnitude in a spatially uniform domain. This figure is not applicable to the experimental regime of the paper (or at least the authors have failed to show that this is the case).

Figure 1c is an interesting and worthwhile experiment (uniformly activate by optogenetics all excitatory neurons in a region of visual cortex) but is especially unclear how this is connected to a Turing or other pattern-forming instability: what is the nature of the cortex just before the uniform optical stimulus is turned on? The authors seem to be assuming that they are perturbing a uniform medium and so are trying to influence the infinitesimal exponentially growing modes of a traditional Turing instability. But the ferret's cortex already has undergone some pattern formation so panels c, d, and e of Figure 1 correspond to some difficult nonlinear

already saturated regime that has no obvious relation to a Turing instability.

The only way I can make sense of this, which is consistent with Figures 2 and 3, is that the authors are optogenetically stimulating an already existing nonlinear pattern of cortex. This is, again, interesting to explore but is an entirely different problem than what the paper claims to be interested in, which is to explain why a cellular pattern forms in the first place in visual cortex.

In any case, the introduction of the paper and the discussions of Figures 1-3 are deeply confusing because the authors are not being clear about what they mean by a Turing mechanism. They are definitely not discussing a Turing instability and not explaining how or why a cellular pattern arose in the first place in immature ferret cortex. Instead, they are investigating what happens when they optically activate just the excitatory neurons in a part of cortex that has already undergone substantial pattern formation.

2. A next serious weakness of this paper is that the authors do not discuss development nor self-organization so the title of this paper and much of the introduction and conclusions need to be rewritten. The authors nowhere present data regarding how the properties of the visual cortex are changing over time, how the patterns are self-organizing over time. All that is being studied is how preexisting poorly characterized neural activity patterns at a certain time during development are responding to optogenetic perturbations. Again, interesting and worthwhile but not what the authors claim they are studying.

A future revised version of this paper would be strengthened if the authors could present results similar to Figures 2-3 over different developmental periods. The authors should also present, as a control, results for an adult ferret (say in a completely dark chamber so its eyes are not active and with its cortex also optogenetically activated in the same way). And maybe another useful control would be to repeat these experiments in preborn ferrets for which the optic nerves have been cut, so pattern-formation cues from retinal waves have been turned off.

3. A third serious weakness of the paper is that the authors fail to discuss how optogenetic stimulation of just the excitatory neurons in visual cortex is related to the kind of neural activity that arises in unperturbed ferrets from retinal waves or from the eyes when the eyes are open. It is not obvious to this referee why a spatially uniform optogenetic activation, or a spatially patterned activation like Fig 3(a), has anything to do with the activation that would occur if the ferret were looking at such a pattern with its eyes. This failure to discuss how optogenetic stimulation is or is not related to visual stimulation with the eyes weakens the paper since the optogenetic stimulation could be a highly abnormal way to stimulate visual cortex.
4. A related issue, similar to the previous one, is that, in Figures 2 and 3, the authors are applying time-dependent optogenetic stimulations. That is, they are looking at a periodically driven piece of cortex, and the repetition time, once every five seconds in Figure 3a, is not so long in terms of synaptic time scales (10-30 ms) or possible times for neurons to habituate or for synapses to undergo plastic changes. Using

time-dependent perturbations of a nonlinear disordered spatial pattern is interesting but far from being relevant for Turing instabilities or Turing mechanisms.

I also include the following miscellaneous comments that are less important but hopefully helpful if the authors revise their paper or decide to submit their paper elsewhere.

1. Several places in the paper, including at the beginning of the abstract, the authors mention that primates and carnivores (which a ferret is) have cortical activity with a distributed modular pattern. This is a puzzling statement on two accounts. First, why do carnivores have modular cortical patterns but not non-carnivores of comparable body and brain size (say wolves versus deer)? Because only carnivores have foveated eyes and perhaps binocular vision?

And second, why don't any other larger-brain animals besides primates and carnivores have modular structure, say elephants, seals, and cetaceans? Is anything known about modularity for birds (which can have foveated eyes, like hawks) and for cephalopods, which can have large foveated eyes (bigger than a human head for giant squids) and presumably might have evolved to also have modular structures in the visual parts of their brains. In any case, the mention of just these two cases is strange and doesn't sound quite right.

A third point is that there are modular (cellular) patterns in many places in the brain, e.g., the author do not mention cortical columns (the oldest and most famous example) or barrel cortices in mice, and even in smaller animals like flies, whose ommatidia lead to spatially periodic vertical organizations of neurons. So what is special about visual cortex versus these other regions? The lack of discussion of how the present paper relates to other cortical modular structures is unfortunate.

Brain development is really a subbranch of biological development and there are lots of examples of cellular patterns forming outside the brain (stripes and spots on the skin, fingerprints, the segmented structure of a spinal cord, formation of five fingers on a hand, etc) and some of these mechanisms are known to be related to Turing instabilities and some involve non-Turing mechanisms like cells that detect absolute concentrations of some morphogen forming a spatial gradient and then the cell undergoes gene activation and specialization when it detects a concentration above or below some threshold (a so-called French-flag model of embryonic patterning). So the really interesting question the authors should address and discuss is to what extent there are pattern formation mechanisms occurring in brains that do not occur outside the brain. It is remarkable that the authors do not cite any references about biological pattern formation since the Turing instability is just one of many known pattern formation mechanisms.

2. The authors mention way too briefly that "scaffolding" is an alternative theoretical exploration to a Turing instability for explaining modular activity patterns. In a revised paper, the authors should not assume that the readers know what scaffolding is and should add a few sentences of explanation and context.

influenced by lateral boundary conditions that define the boundary of the domain, e.g, boundary conditions can suppress patterns, favor a unique pattern, cause time-dependent patterns, etc. The authors completely ignore any consequences of the finiteness of the cortex and any possible influence of boundaries (say caused by changes in the properties of the neurons as one goes from one part of cortex to another). Experiments at some point will be needed to determine if boundaries are having any influence.

3. The discussion of noise in the paper is too brief and unsatisfying. It would be useful to state briefly what are some sources of noise and try to estimate the magnitudes of these sources. Most pattern formation discussions assume that noise is negligible once a pattern has become nonlinearly saturated but this may not be true for the cortex, especially if synaptic plasticity causes neural interconnections to keep changing in strength.
4. Figure 2 (and, to some extent, the other figures) are too complex, with too many panels and with too many details included that are never discussed or discussed so briefly that only a few experts know what they mean. (E.g., the meaning of the colors in Fig. 1a, the green and purple wedges in Fig 1b are not mentioned, Fig. 2d is not discussed well in the caption or text and makes little sense, the dice in Fig 1 aren't explain in the caption and not mentioned until long after Figure 1 appears so are just confusing, so just drop these details or figures). I would strongly recommend the authors to reduce the number of panels and details per panel as much as possible. It is the quality of the insight and of the logic that will make this paper interesting and significant, not trying to include every measured or calculated detail, which overwhelms most readers. Any detail shown in a figure should be explained in enough detail to justify to the reader why the detail was included, otherwise, don't include the detail or panel.

In summary, this paper has many substantial flaws that make it unacceptable for publication in its present form and will require a major rewriting to become acceptable. By far the greatest weakness is a lack of clarity about what is a “Turing mechanism” and how the current experiments relate to a Turing mechanism. The paper, in fact, is not studying development nor self-organization nor a Turing mechanism but how already structured visual cortex responds to a variety of time-dependent optogenetic stimulations of unknown relation to how the same cortical neurons would respond to known visual stimuli. While the data are interesting and worth publishing, the issues are quite different than what the present paper claims, as an explanation of self organization.

Reviewer #2 (Remarks to the Author):

Mulholland and colleagues conducted simultaneous calcium imaging and optogenetics in the developing ferret visual cortex to test a theoretical model of cortical circuit development. They characterized response of the visual cortex to various spatial patterns of optogenetic stimulation and showed that the experimental results matches with simulations based on the Turing mechanism. They further manipulated activity of the Thalamus and the cortex to show that the optogenetic response patterns were mainly created within the cortex.

While many studies use the mouse as a model system to study cortical circuit development, the mouse brain lacks some important features such as functional columns that are widely observed in other mammalian species including humans. In this study, authors nicely combined cutting edge experimental techniques with sophisticated computational models in the ferret visual cortex and characterized early developmental process of columnar functional circuits. Overall, I think this study is a very important step toward understanding the development of columnar cortical circuits.

Below are my comments.

Major comment

It was not clear to me how the spatial patterns of optogenetic stimulation affected "self-organization" of cortical activity.

(A) Do spatially structured optogenetic stimulation produce novel spatial patterns not observed in spontaneous activity?

Or, (B) do spatially structured optogenetic stimulations just increase occurrence of subsets of spontaneous activity patterns? (similarly to uniform stimulation which increases the occurrence of all spontaneous patterns)

If the latter case (B) were true, is it consistent with the Turing model?

I think the authors can check this point, both for animal experiments and computational simulations, by comparing spatial patterns of *individual* optogenetic events (in Figures 3&4) with spatial patterns of spontaneous activity (or uniform stimulation).

If the case (B) were true and not consistent with the current Turing model, the authors should weaken the statement that cortical activity patterns are self-organized, throughout the manuscript.

Minor comments

1) When I first read the manuscript, I did not notice that the optogenetically evoked activity patterns are highpass filtered and mean subtracted. To be able to assess relative strength of filtered and non-filtered activity patterns, please add non-filtered & non-mean subtracted examples, such as an example shown in Fig2d, in Fig3 and Fig4.

2) lines887-899 Authors stated "For simplicity, we use a linear rate network model with strong recurrent connections close to the critical value ... the maximum eigenvalue of the connectivity matrix is equal to 0.99".

I am curious how sensitive the overall simulation results are to the specific choice of the eigenvalue. Also, it would be nice if the authors could discuss biological plausibility of the developing cortex being at the critical state.

3) In Fig3f, the values of similarity between individual events and optogenetics spatial patterns seem quite low, hence it is not clear to what extent individual events reflect optogenetics spatial patterns. I think this is partly because only a small fraction of pixels are activated/deactivated in each event (in highpass filtered images). Can the authors calculate correlations using only those pixels that showed large activity change in each events (e.g. use pixels with $|z| > 2$)?

4) It is not mandatory but it would be nice if the authors could present a data showing that GCaMP and ChrimsonR are co-expressed in single neurons (which I think is the assumption by the authors).

5) Y axis in Fig5g and Fig5h should be the same.

5) In Fig5i, median of the control is somewhere around 0.2. But in Fig5j, there's no control experiment around Modularity~0.2. Please check.

Reviewer #3 (Remarks to the Author):

This is an impressive manuscript from Mulholland, Kaschube and Smith, which combines widefield fluorescent calcium imaging with optogenetic stimulation to show that modular responses emerge from cortical interactions in the developing ferret visual cortex (V1). This work is motivated by the prior finding that modular activity patterns at the spatial scale of cortical columns are present in the spontaneous activity of visual cortex prior to eye opening. They propose a neural implementation which

requires recurrent excitation of neighboring neurons -which form a module- combined with longer range inhibition between modules. Such a model generates periodic groups of co-activated modules which resemble the periodic functional maps observed in primate and carnivore neocortex. The authors describe 3 predictions of the Turing model, each of which they test by optogenetically stimulating ferret V1 prior to eye opening. First, they show that non-specific activation of V1 with optostim produces modular and periodic patterns of neural activation, in line with the Turing prediction. Next, they stimulated V1 with random patterns at the characteristic wavelength of cortical modules observed during spontaneous activity. Indeed the trial-averaged activation patterns in V1 strongly resembled the spatial structure of the optostim. To further test the validity of this candidate mechanism, they presented optostim patterns at spatial wavelengths which differed from the characteristic wavelength of cortical modules observed during spontaneous activity. Their model predicts that inputs away from the characteristic wavelength will produce a response at an intermediate wavelength. They show that indeed, the spatial wavelength of cortical activation following patterned stimulation lies at a value between the input and characteristic wavelengths, and that this effect is present when input wavelength is both narrower and broader than the characteristic wavelength. This final point is quite valuable, as it shows that modular patterns of cortical activation do not simply reflect the spatial pattern of inputs, nor are they confined to a single wavelength regardless of input.

An alternative hypothesis for modular responses is that the structure of the feedforward activity is essential. The authors reject this hypothesis by showing that modular activation following non-specific optostim persists when LGN is silenced with Muscimol.

Finally, the authors compare the optostim-evoked activation patterns to the spontaneous patterns which they had previously observed in cortex. They find that these patterns are highly correlated in their spatial arrangement and can be accounted for by the same low-dimensional principal component representation. This result indicates that these patterns emerge from a common synaptic source- an important finding in support of their hypothesis that a Turing-like organization seeds the formation of modular, periodic maps in cortex.

This is a very strong manuscript. The authors lay out a clear hypothesis – that the Turing-like mechanism describes the population-level organization of ferret cortex prior to eye opening. They make testable predictions of how a neural population organized in this way would respond to different patterns of input. They show that each of these predictions is met in-vivo. This work represents a significant advance in our understanding of cortical organization early in development. It also constrains models of functional maps in cortex by indicating that columnar organization is an emergent property of the structure of intracortical connections present prior to visual experience, rather than depending on selective pooling of peripheral afferents.

I have only one critical comment for this work. The manuscript seeks to describe the neural mechanisms which generates modular organization in the absence of structured visual input. And yet there may be

many circuit mechanisms that implement this computational model. Critically some of the circuit models that the authors cite, for example, the attractor models of Sompolinsky, exhibit dynamics that do not seem to match cortical networks. Are there any aspects of the cortical dynamics from the present recordings that could make constraints on the neural implementation of this Turing-like phenomenon?

Reviewer 1

1. The paper has serious conceptual flaws. One is that the authors never clearly state what they mean by a Turing mechanism so it is difficult to determine what they are claiming in their paper.

The problem is that Turing's original calculation in 1953 has been now discussed and somewhat reinterpreted by such a big community of researchers (physicists, mathematicians, chemists, biologists, engineers, and others) that the concept of a "Turing mechanism" has become so broad as to be useless unless authors state exactly what they are talking about, which the authors do not in this paper.

Turing's original calculation was rather narrow and dealt only with the linear instability (growth of infinitesimal perturbations of all different wavelengths) associated with an infinitely wide continuous medium that had no boundary conditions imposed. (Turing discussed mainly two reacting and diffusing chemicals, although he did discuss briefly a discrete medium consisting of a periodic ring of cells whose sizes were tiny compared to the wavelength of the instability). Turing showed by his pioneering theoretical calculations that, under certain circumstances (such as varying the ratio of the diffusion constants of two chemicals), a spatially uniform medium could become unstable to exponentially growing spatially sinusoidal modes with a narrow distribution of wavelengths centered on the mode with the fastest growing rate (the so-called critical wavelength of the linear instability).

But a key point is that Turing only discussed the onset of a linear instability and did not discuss (because the mathematics was too hard at the time) what the linearly growing modes would evolve into when the nonlinearities of the evolution equations eventually cause the amplitudes of the growing modes to saturate to have finite values. This saturated nonlinear regime is a new and different problem in pattern formation that Turing did not address and many researchers, including the authors, unfortunately and incorrectly, call the nonlinear saturated pattern a Turing pattern or the result of a Turing mechanism (words used in the current paper).

This is an extremely important point for the authors to clarify in their paper. The authors at no time discuss the linear instability of an initially approximately-uniform patternless piece of cortex and so they never are in a position to observe a classical Turing instability. Instead they are using optogenetics to optically perturb a portion of the visual cortex of a still developing ferret after the cortex already has developed an intricate inhomogeneous spatial structure that is not well characterized. (Local excitation and longer-range lateral inhibition indeed exist but this is an extremely crude qualitative observation since the anatomical and physiological details have not been measured of the developing cortex and so are poorly characterized.) That is, the authors are reporting results for a medium (visual cortex) whose spatial symmetry is already broken (is nonuniform). While the results are still interesting, the results lie outside of what Turing has discussed and, in fact, lies outside most pattern formation theory since the nonlinear saturation of unstable modes in a finite disordered inhomogeneous medium or the effects of order-one perturbations of a nonlinear saturated medium is a difficult and only partially understood problem.

So Figure 1 of the paper is greatly misleading. Fig. 1b shows, but fails to explain adequately, the growth rates $\sigma(\lambda)$ for infinitely-wide sinusoidal modes of infinitesimal magnitude in a spatially uniform domain. This figure is not applicable to the experimental regime of the paper (or at least the authors have failed to show that this is the case).

Figure 1c is an interesting and worthwhile experiment (uniformly activate by optogenetics all excitatory neurons in a region of visual cortex) but is especially unclear how this is connected to a Turing or other pattern-forming instability: what is the nature of the cortex just before the uniform optical stimulus is turned on? The authors seem to be assuming that they are perturbing a uniform medium and so are trying to influence the infinitesimal exponentially growing modes of a traditional Turing instability. But the ferret's cortex already has undergone some pattern formation so panels c, d, and e of Figure 1 correspond to some difficult nonlinear already saturated regime that has no obvious relation to a Turing instability.

The only way I can make sense of this, which is consistent with Figures 2 and 3, is that the authors are optogenetically stimulating an already existing nonlinear pattern of cortex. This is, again, interesting to explore but is an entirely different problem than what the paper claims to be interested in, which is to explain why a cellular pattern forms in the first place in visual cortex.

In any case, the introduction of the paper and the discussions of Figures 1-3 are deeply confusing because the authors are not being clear about what they mean by a Turing mechanism. They are definitely not discussing a Turing instability and not explaining how or why a cellular pattern arose in the first place in immature ferret cortex. Instead, they are investigating what happens when they optically activate just the excitatory neurons in a part of cortex that has already undergone substantial pattern formation.

We thank the reviewer for their thoughtful feedback. The reviewer raises one major point which highlights several aspects in which the communication of our results was vague, confusing, or in need of reframing. Below, we respond to this feedback at length, providing a clearer explanation of our work, before addressing the other points raised by the reviewer.

In order to address the first major concern, below we first clarify a few points of potential misinterpretation of our study ("*Points of clarification*", before providing a more detailed explanation of the mechanism that we test, based on a model ("*Pattern formation in LE/LI systems*"), as well as a justification of our experiments. Finally we summarize how we have addressed the above suggestions from this reviewer in the revised version of the manuscript ("*Summary of changes*").

Points of clarification

1. Self-organization, broadly speaking, refers to a process where global order arises spontaneously from the local interactions between a system's parts. In our paper, we do not study the self-organization of neural circuits (or neuronal selectivity patterns, such as e.g. ocular dominance maps) over the course of development, i.e. on time scales hours to weeks, which is the time scale of neuronal plasticity and synaptic changes. Instead, our goal is to test long-standing models for the self-organization of neural activity patterns, on the time scale of milliseconds to seconds, the time scale of propagation of activity within the cortical network. We regret that our introduction was not sufficiently clear about this important distinction and we have revised it accordingly.
2. The principal aim of this paper was to test predictions of a broad class of models that have put forward the hypothesis that modular patterns of cortical activity form through a process of self-organization of neural activity resulting from recurrent local excitation and lateral inhibition (also sometimes referred to in the literature as 'Mexican hat' or difference of Gaussians). We use the term 'modular' to refer to activity patterns that consist of local domains of active neurons with roughly regular spacing, such that the pattern is dominated by a single characteristic spatial wavelength. Virtually all spatially extended spontaneous activity in the early developing ferret visual cortex is modular [30]. Recurrent network interactions composed of effective local excitation and lateral inhibition can selectively amplify such modular patterns of activity and are, therefore, a candidate mechanism for modular activity in the early cortex. We describe this mechanism in more detail below based on a concrete representative of such a model (see "*Pattern formation in LE/LI systems*").
3. By using the term 'Turing mechanism' to refer to the mechanism of pattern formation in these models, we followed the interpretation by Kondo and Miura stating in their review article in 2010 in *Science* [17] (also cited in our previous manuscript): *Gierer and Meinhardt [21, 20] showed that a system needs only to include a network that combines "a short-range positive feedback with a long-range negative feedback" to generate a Turing pattern. This is now accepted as the basic requirement for Turing pattern formation [37, 21].* According to this view, the term 'Turing mechanism' is not restricted to a specific time scale or implies a pattern formation process over the course of development, but is also applicable to pattern formation during neural activation. Notably, this use of the phrase 'Turing mechanism' in this context to describe pattern formation in cortical network activity has been used previously in the literature [3]. Moreover, the class of models that we sought to test in our study clearly satisfies the 'basic requirement' of Kondo and Miura.

We do, however, acknowledge the criticism by this reviewer that the terms 'Turing mechanism' and 'Turing patterns' are used inconsistently in the literature and may thus be confusing. We therefore revised our manuscript to now avoid 'Turing patterns', and now refer to 'local excitation and lateral inhibition' (abbreviated with 'LE/LI') instead of 'Turing mechanism'. We still cite Turing's original work (and the review by Kondo and Miura) to acknowledge his original contribution.

4. This reviewer seems to assume that with our optogenetic stimulation we elicit activity patterns that are "already existing" at this early stage in development. However, it is important to note that, currently, there is no experimental support for such an assumption. For instance, in the literature, clustered long-range horizontal connections have repeatedly been suggested to impose its modular structure onto cortical activity [29, 11]. These clustered connections were observed to align to the layout of orientation preference columns in the mature cortex, but these clustered connections are not present yet at the considered early stage in development [9]. Instead of reflecting a specifically structured intracortical network, modular cortical activity could, in principle, be brought in from outside, for instance by the structure of feed-forward inputs. However, our pharmacology experiments argue strongly against this possibility, indicating that the cortical network is necessary and sufficient for generating modular activity, in line with our previous observations that spontaneous modular activity in cortex persists after blockade of all activity in the retina, and even after silencing LGN [30].
5. It is precisely this assumption of already existing modular patterns at this early age that we are challenging with our experiments. In our study, we are testing the alternative possibility that the modular structure of activity is generated spontaneously *at the time of cortical activation* through recurrent network interactions of the type LE/LI. Such network interaction amplifies patterns around a characteristic spatial wavelength, thereby inducing some degree of global ordering, but as such does not impose a spatial phase of the pattern. Consistent with this possibility, not only do we find that uniform activation drives modular patterns of activity (Fig. 2) – and with a variable layout of active domains across repeated stimulations – but also that patterned activation with a wavelength near the characteristic wavelength of spontaneous activity results in activity patterns that are novel, in the sense that they deviate from the patterns of uniform stimulation and also from those naturally occurring during spontaneous activity at this age, as we now show in the revised Fig. 6. These results argue that our optogenetic stimulation does not merely elicit "already existing" activity patterns.
6. Prior to uniform optogenetic stimulation, the cortex tends to be quiescent and the frequency of spontaneous events in our experiments tends to be low (as seen in Fig 2b-d, and supplemental movie 1). Therefore, frequently the state of neural activity in the cortex prior to our stimulation is low and uniform. When we optogenetically stimulate, with a sufficiently high intensity, we see an increase in calcium activity across our field of view and a modular pattern emerges, consistent with pattern selection at the timescale of neural activation. For low intensity, the activity is weaker and often remains close to uniform (Fig. S3), indicating that modular activity requires stimulation above some threshold intensity. Note that, in order to avoid having our results be contaminated by spontaneous activity, in our quantification of modularity and wavelength we excluded trials where cortical activity was present within 1 second prior to opto-stimulation.
7. We agree with this reviewer that at the developmental age our experiments are performed, the cortical circuitry, while still mostly short-range, likely exhibits some degree of heterogeneity across cortical space. However, neither is it evident that this heterogeneity could generate the modular activity we observe, nor is some mild degree of heterogeneity inconsistent with a pattern forming mechanism based on LE/LI. As we showed previously [30], the main effect of a mild degree of heterogeneity is a reduction in the diversity of activity patterns, while individual patterns maintain their modular structure. Importantly, the three predictions we test in the present study (Fig. 1) hold also in the presence of such heterogeneously perturbed LE/LI interactions (see Fig. 4a-c and Fig. S5 for the heterogeneous model and Fig. S12 for the homogeneous model).
8. Selective amplification through local excitation and lateral inhibition (LE/LI) appears to be a robust mechanism for producing modular patterns of activity, not only in a supercritical, but also in a slightly subcritical regime, in which there is no pattern instability, but modular patterns can still form if the

inputs contain a random (e.g. white noise) component, as we show in our manuscript (Figs. S5, S12). Both the connectivity and the various noise sources remain poorly characterized in the early cortex, but the fact that this mechanism is so effective, irrespective of many model details, makes it a plausible candidate mechanism for modular cortical activity.

Pattern formation in LE/LI systems

In the following, we will discuss a concrete example of a model in order to explain more clearly the mechanism we are testing and the claims we are making in the paper. This model describes the firing rate u_E of a single (infinitely large) excitatory neural population in two dimensions (a ‘neural field’). The lateral interactions are assumed to be identical for each neuron and the external input drive is assumed to be constant in space and in time. (Later, also other forms of input are considered.)

The firing rate u_E of this excitatory neural population is governed by

$$\tau_E \frac{du_E(\mathbf{x}, t)}{dt} = -u_E(\mathbf{x}, t) + \left[\int d\mathbf{y} (a_E M_E(|\mathbf{x} - \mathbf{y}|) u_E(\mathbf{y}, t) - a_I M_I(|\mathbf{x} - \mathbf{y}|) u_E(\mathbf{y}, t)) + J \right]_+ \quad (1)$$

where τ_E is an effective time constant related to the membrane time constant (several tens of milliseconds), $J > 0$ is an external input assumed to be constant in space and time, $a_E > 0$ and $a_I > 0$ control the overall strengths of the excitatory and inhibitory recurrent connections M_E and M_I , respectively, and $[x]_+$ denotes rectification ($x = 0$ for $x \leq 0$, $x = x$ for $x > 0$). The shape of these connections is assumed to be isotropic and to fall off as a Gaussian function

$$M_K(|\mathbf{x} - \mathbf{y}|) = \frac{1}{2\pi\sigma_K^2} \exp\left(-\frac{|\mathbf{x} - \mathbf{y}|^2}{2\sigma_K^2}\right) \quad (2)$$

with $K = \{E, I\}$. For $\sigma_I > \sigma_E$ the connectivity is of the type LE/LI. Fig. 2, left, displays this type of connectivity and illustrates the case of LE/LI.

A spatially homogeneous fixed point solution of Eq. 1 satisfies

$$0 = -\bar{u}_E + [a_E \bar{u}_E - a_I \bar{u}_E + J]_+$$

We are looking for solutions with a positive firing rate, $\bar{u}_E > 0$. For such a solution the rectification can be dropped and the solution is then given by

$$\bar{u}_E = J \frac{1}{1 - a_E + a_I}$$

assuming $a_E < a_I + 1$.

Next, we seek to study the conditions under which this uniform solution \bar{u}_E is unstable. To this end, we insert

$$u_E(\mathbf{x}, t) = \bar{u}_E + \epsilon w_E(\mathbf{x}, t)$$

with $0 < \epsilon \ll 1$ into our full model equation (Eq. 1) to study the growth rates of infinitesimal perturbations $w_E(\mathbf{x}, t)$ around this uniform solution. Since $\bar{u}_E > 0$, the rectification can be dropped resulting in a linear equation

$$\tau_E \frac{dw_E(\mathbf{x}, t)}{dt} = -w_E(\mathbf{x}, t) + \int d\mathbf{y} (a_E M_E(|\mathbf{x} - \mathbf{y}|) w_E(\mathbf{y}, t) - a_I M_I(|\mathbf{x} - \mathbf{y}|) w_E(\mathbf{y}, t)) \quad (3)$$

describing the evolution of the perturbation $w_E(\mathbf{x}, t)$. As the connectivities M_E and M_I are the same at each location \mathbf{x} , plane waves $\sim \exp(-i\mathbf{k}\mathbf{x})$ are eigenfunctions of the right-hand side of Eq. (3). Furthermore, as the connectivities depend only on the distance between locations \mathbf{x} and \mathbf{y} the corresponding eigenvalue spectrum depends only on the absolute value of the wavenumber $k = |\mathbf{k}|$ and is given by

$$\lambda(k) = -1 + a_E \exp\left(-\frac{1}{2}\sigma_E^2 k^2\right) - a_I \exp\left(\frac{1}{2}\sigma_I^2 k^2\right) \quad (4)$$

For $\sigma_E < \sigma_I$ (LE/LI) and a_I and a_E sufficiently large, this spectrum has a single peak, slightly above zero, at a finite wavenumber k_0 , while eigenvalues approach -1 for wavenumbers much smaller or larger than k_0 (Fig. 2, right, solid line). Thus, plane waves around k_0 are unstable and grow exponentially, while those with a small or large wavenumber rapidly decay to zero, implying that a spatial pattern grows that exhibits a characteristic wavelength $\Lambda = 2\pi/k_0$. The growth of this pattern eventually saturates due to the rectification nonlinearity in the full model (Eq. 1) and the dynamics converges to a hexagonal pattern [30].

Figure 1: Left: Local excitation and lateral inhibition (LE/LI) in the model (Eq. 1) arising from a Gaussian shaped connectivity (Eq. 2) that is wider for inhibition than for excitation. Right: The eigenvalues (Eq. 4) exhibit a single maximum at a finite wavenumber k_0 with values near zero, whereas eigenvalues for low and high wavenumber are strongly negative close to -1. If the maximum is above zero (solid line), plane waves $\sim \exp(-i\mathbf{k}\mathbf{x})$ with a k around this wavenumber k_0 will grow exponentially according to Eq. 3, while those with low or high wavenumber will rapidly decay to zero, resulting in the growth of a pattern dominated by the characteristic wavelength $\Lambda = 2\pi/k_0$. If the maximum is slightly below zero (dashed line), plane waves around k_0 will decay to zero, but much slower than those for small and large k resulting in the selective amplification of a pattern dominated by $\Lambda = 2\pi/k_0$ in case the network is activated by external input. Parameter values: $\sigma_E = 1$, $\sigma_I = 2$ and $a_E = a_I = 2.3$ (solid), $a_E = a_I = 1.9$ (dashed).

The formation of an activity pattern in this model is a process of self-organization, as the pattern that is forming is not brought in from outside (given the external input is uniform in space and time), but arises from the recurrent interactions via a dynamic instability of the homogeneous solution. Whereas the dynamics is symmetric with respect to continuous translation and rotation, a given solution in the form of a hexagonal pattern exhibits a much more reduced symmetry. The recurrent interactions determine the wavelength, but not the global spatial phase and orientation of the pattern that forms, which also depends on the initial conditions and, potentially, on other factors, such as noise or boundary conditions. In these respects, the dynamics considered here share several aspects with other pattern forming systems such as, for instance, Rayleigh-Bénard convection.

A simple extension of the model in Eq. 1 is to include a threshold Θ under the nonlinearity, equivalent to an input $J - \Theta$. Such type of threshold is frequently used with rate units and can be linked to the voltage threshold in spiking neuron models. In the presence of such a threshold, modular patterns could only start forming once the input J exceeds this threshold. Interestingly, by varying the laser power of our uniform stimulation, we observe a nonlinear increase in the degree of modularity as a function of laser power that appears consistent with such threshold (SI Fig. 3).

It is important to emphasize that if the system Eq. 1 is driven by an input that contains a broad (e.g. white) noise component, modular patterns will also form if the peak of the spectrum remains slightly below

zero (as in Fig. 1, right, dashed curve). The two regimes (slightly below or above this critical point) differ somewhat with respect to the final patterns they produce, as these are shaped differently by the nonlinear terms of the dynamics. However, in both regimes there is a selective amplification of modes close to the critical spatial frequency due to the large difference in growth rate relative to frequencies that are considerably smaller or larger. Since our manuscript seeks to test generic predictions of this basic mechanism for the generation of modular structure – and not addressing the nonlinear dynamics close to pattern saturation – we focused on the simpler of these two regimes, i.e. the subcritical one, to derive model predictions.

Fig. 1 in the manuscript then proposes to test three predictions of this mechanism. First, spatially uniform stimulation (which in the cortex is expected to also have significant noise components (see below)) should result in modular activity patterns that vary from stimulation to stimulation. Second, spatially structured stimulation with the same wavelength as patterns that naturally occur (i.e. spontaneous activity) should be able to bias the pattern, since the mechanism described above primarily constrains the spatial wavelength, but not the overall layout of the pattern. Third, spatially structured stimulation with a deviating wavelength should result into a dynamic compromise between the intrinsic (critical) wavelength and the wavelength of the stimulation, biasing the wavelength of the resulting patterns towards the stimulation wavelength.

A more realistic model of the early visual cortex should assume the connectivity exhibits some degree of variation across cortical locations. The amount of heterogeneity may be lower compared to the mature cortex when the network is fully refined and the intricate array of clustered long-range horizontal connections established. Moreover, the effective heterogeneity may be weaker on the columnar level that we are modelling here. Nevertheless, some degree of heterogeneity is expected, and thus we studied our network model in both a homogeneous and a moderately heterogeneous regime (by randomly perturbing the Gaussian connectivity kernels, see Methods) and we obtained consistent results with modular patterns in both cases and quantitatively similar predictions for the phenomena investigated in Figs. 2-4 (SI Figs. 5,12).

Moreover, in the cortex, there is typically a substantial amount of seemingly random influences ('noise') affecting the propagation of neural activity within the network. These include, for instance, synaptic failure and fluctuating neural excitability. As a simple model of this inherent stochasticity, we fed spatially uncorrelated noise into the network, in addition to the input J . Even for strong noise (having a standard deviation on the order of J), activity patterns remained modular. Moreover, in this regime the model showed a close match to the data for the predictions tested in Figs. 2-4.

Summary of changes

We acknowledge the potential ambiguity around our use of the term 'Turing mechanism' and the potential for confusion surrounding the mechanism we seek to test. We have extensively revised our manuscript to provide additional clarity and specificity, and to better explain the proposed mechanism and the rationale for our experiments.

Additionally, throughout our revised manuscript, we have re-framed our rationale and discussion around specifically testing models that use LE/LI, and we de-emphasized the connection between our results and the Turing mechanism, while still placing LE/LI within the broader context of mechanisms derived from Turing's work. We feel that this makes the interpretation of our experiments more clear, while also contextualizing these results with broader theories of pattern formation.

2. A next serious weakness of this paper is that the authors do not discuss development nor self-organization so the title of this paper and much of the introduction and conclusions need to be rewritten. The authors nowhere present data regarding how the properties of the visual cortex are changing over time, how the patterns are self-organizing over time. All that is being studied is how preexisting poorly characterized neural activity patterns at a certain time during development are responding to optogenetic perturbations. Again, interesting and worthwhile but not what the authors claim they are studying.

A future revised version of this paper would be strengthened if the authors could present results similar to Figures 2-3 over different developmental periods. The authors should also present, as a control, results

for an adult ferret (say in a completely dark chamber so its eyes are not active and with its cortex also optogenetically activated in the same way). And maybe another useful control would be to repeat these experiments in preborn ferrets for which the optic nerves have been cut, so pattern-formation cues from retinal waves have been turned off.

We are grateful to the reviewer for pointing out this potential confusion, which also stimulated parts of our response to Point 1 above. As described in detail in our response to Point 1, we aimed to test whether the patterns of activity that are observed in the immature ferret brain prior to eye opening arose through a mechanism that self-organizes activity – that is, large-scale ordered patterns arise through local intracortical circuit interactions at the time the cortex is activated, rather than being preexisting in cortex or inherited from other brain regions through already structured inputs. We demonstrated that direct uniform stimulation of the cortex leads to spatially structured output patterns, even in the absence of feedforward input from the LGN. Additionally, we showed that blocking intracortical glutamatergic activity prevented modular patterns from forming. Together, this supports that the mechanism that gives rise to these patterns likely resides in intracortical circuits. Moreover, these networks shape cortical activity from uniform stimulation, resulting in patterns that are similar to those that occur endogenously. The fact that these uniform opto-evoked patterns are similar to patterns of endogenous activity does not necessarily imply that these patterns are preexisting. The more plausible interpretation (making less assumptions), however, is that these endogenous patterns form in response to uniform or noisy input through a similar spontaneous ordering process as the opto-evoked patterns.

Importantly, spatially structured stimuli can bias the patterns of evoked responses, such that the resulting patterns deviate from those endogenous patterns. In the revised manuscript we now show this explicitly in our revised Fig. 6. This result argues against the hypothesis that the evoked patterns preexist. While biased towards the stimulation pattern, the resulting activity patterns also do not exactly match the stimulation patterns, but rather still show some overlap with the endogenous patterns, suggesting that the activity arises via dynamically compromising network interactions and the external input (Fig. 6f-h). This is consistent with our model network with LE/LI interactions when driven by structured noisy inputs (Fig. S11). A similar dynamic compromise can be seen for stimulation patterns with a wavelength that deviates from the characteristic wavelength of the system (Fig. 4).

Regarding development, we performed these experiments in young ferret visual cortex in order to address questions about whether modular activity can arise from short-range network interactions at synaptic timescales. At no point did we intend to draw conclusions on how these network properties change over time, which though a fascinating question is beyond the scope of this single manuscript. Though the Turing mechanism has frequently been applied to computational models of functional map development which incorporate learning rules for how these networks could be trained to encode specific visual features over time, such as orientation selectivity or ocular dominance (e.g. [35, 32, 15]), it has also been used to model the emergence of spontaneous neural activity patterns (e.g. [10, 3, 30]). In our revisions of the manuscript, we clarify these related but separate ideas, and only in the discussion relate our findings in the longer-term implications of their role in the development of functional maps for visual features.

Though previous work has demonstrated that spontaneous activity in developing V1 after retinal wave blockade on the day of imaging is still modular [30], and thus retinal activity does not seem required for large-scale modular patterns to occur, it is unknown what effect retinal blockade from birth would have on functional activity in V1 prior to eye opening. Repeating our experiments in enucleated ferrets would again address questions of development and its dependence on visual experience, which, while interesting, is not the aim of our study.

In our revised manuscript, together with the changes noted above, we also highlight that our hypothesis and results address the generation of modular activity patterns on synaptic (i.e. neural activity) and not developmental timescales. We have also revised the title of our manuscript to "Self-organization of modular activity in immature cortical networks" to further clarify that we are examining this question at a single point in early cortical network development.

3. A third serious weakness of the paper is that the authors fail to discuss how optogenetic stimulation of just the excitatory neurons in visual cortex is related to the kind of neural activity that arises in unperturbed ferrets from retinal waves or from the eyes when the eyes are open. It is not obvious to this referee why a spatially uniform optogenetic activation, or a spatially patterned activation like Fig 3(a), has anything to do with the activation that would occur if the ferret were looking at such a pattern with its eyes. This failure to discuss how optogenetic stimulation is or is not related to visual stimulation with the eyes weakens the paper since the optogenetic stimulation could be a highly abnormal way to stimulate visual cortex.

Here, the reviewer highlights an additional area where our explanation of the experimental rationale in our original manuscript was unclear. The reviewer is correct to note that the patterns of incoming activity into visual cortex from feed-forward inputs (driven by either retinal waves or visual stimuli) are potentially unrelated to the patterns of input activation we drive with our optogenetic stimuli. However, the ability to **directly** drive the cortex with arbitrary patterns is precisely one of the key strengths of our experimental design. In our study, we seek to test specific predictions of LE/LI mechanisms by applying specifically designed inputs to the cortex. Studying the response to these (uniform, or patterned) inputs allows us to directly compare our results to model predictions.

Before retinal activity (be it either spontaneous waves or visually-evoked activity driven through closed eyelids) reaches the cortex, it passes through the LGN. As noted above, we conduct our experiments prior to eye-opening, and the precise nature of the retina-thalamus-cortex transformations at this age of development is not well characterized. Therefore it is not possible to directly compare the response of the cortex to arbitrary inputs applied both optogenetically and visually, as it is currently not possible to know what form a retinal input may take when it reaches the cortex.

However, it is possible to draw inferences about the potential nature of these cortical inputs based on their similarity to the patterns of activity evoked through specific optogenetic stimuli. In our original submission, we make two such comparisons: finding a strong similarity between spontaneous events and uniform-opto-evoked events (Fig 6), and a lack of such similarity between visually-driven luminance responses and uniform-opto-evoked events (Fig S7). In our revised manuscript, we now also show that stimulating with spatially structured input patterns can evoke activity patterns with novel components (Fig 6f-h), in a manner consistent with our model (Fig. S11), demonstrating that the cortex can respond to artificial inputs that can reveal key aspects of the underlying network.

In our revised manuscript, we now further elaborate on these findings, and more explicitly explain our rationale for using optogenetic stimulation to directly manipulate input to the cortex.

4. A related issue, similar to the previous one, is that, in Figures 2 and 3, the authors are applying time-dependent optogenetic stimulations. That is, they are looking at a periodically driven piece of cortex, and the repetition time, once every five seconds in Figure 3a, is not so long in terms of synaptic time scales (10-30 ms) or possible times for neurons to habituate or for synapses to undergo plastic changes. Using time-dependent perturbations of a nonlinear disordered spatial pattern is interesting but far from being relevant for Turing instabilities or Turing mechanisms.

As explained in detail above, the time scale on which we are examining the mechanism of pattern formation is the time scale of neural activation (milliseconds to second), for which the applied stimulation paradigm seems well-suited. Our stimuli are spaced sufficiently far apart in time to allow any evoked activity to return to baseline, resulting in a cortex with generally uniform low activity at the time the next stimulus is applied. Our stimulation protocol was not designed to induce changes in synaptic plasticity, but rather probe the response of the cortical network to various stimulation patterns.

In our revised manuscript, as noted above, we more clearly articulate that we are examining activity patterns that emerge on the time-scale of neural activity.

Minor comments:

1. Several places in the paper, including at the beginning of the abstract, the authors mention that primates and carnivores (which a ferret is) have cortical activity with a distributed modular pattern. This is a puzzling statement on two accounts. First, why do carnivores have modular cortical patterns but not non-carnivores of comparable body and brain size (say wolves versus deer)? Because only carnivores have foveated eyes and perhaps binocular vision?

And second, why don't any other larger-brain animals besides primates and carnivores have modular structure, say elephants, seals, and cetaceans? Is anything known about modularity for birds (which can have foveated eyes, like hawks) and for cephalopods, which can have large foveated eyes (bigger than a human head for giant squids) and presumably might have evolved to also have modular structures in the visual parts of their brains. In any case, the mention of just these two cases is strange and doesn't sound quite right.

The question of why some species have modular cortical activity while others do not is a fascinating and longstanding question in comparative neurobiology. Modular cortical patterns have been found in humans, primates, cats, ferrets, tree shrews [6], ungulates [8], and barn owls [19], while Rodents [25, 34], lagomorphs [12] and pigeons [24] have 'salt-and-pepper' cortical architecture. Modularity does not seem to solely be explained by genetic common ancestor [15], body or brain size, or binocular stereopsis. A definitive guiding principle has been elusive, as there seems to be exceptions to every rule. V1 neural density, however, does seem to be a fair predictor of cortical map organization, where creatures with a high density of V1 neurons tend to have modular maps, while lower density V1's tend to be salt and pepper [28, 36]. It has been proposed that modularity is an efficient way to coordinate neurons over large cortical distances by minimizing wiring length [13, 33, 18], which becomes more efficient when there are more cells. This would explain why animals of similar body and brain size but with different evolutionary prioritization for vision end up with different visual cortex architecture. Whether these principles hold true for other larger-brained mammals, such as elephants, seals, and cetaceans, is currently unknown, as it is difficult to assess the functional activity of these species due to access, technical, and ethical reasons, and thus can for now only speculate on their cortical activity organization.

A more detailed discussion of this question has been the subject of several review articles [14, 28]. In our manuscript, we highlight that modular activity is "a hallmark of the primary visual cortex of primates and carnivores", as these groups of species are both the most well-studied in terms of visual cortical function, have relevance to human function (primates) and include the ferret, the model system used in our manuscript (carnivores). In our revised manuscript, in order to acknowledge the large body of work and clarify that we do not believe that these principles are limited to just primates and carnivores, we have included a brief statement in the discussion discussing the reasons why modular activity might have been selected for or could be an epiphenomenon of different evolutionary pressures.

A third point is that there are modular (cellular) patterns in many places in the brain, e.g., the author do not mention cortical columns (the oldest and most famous example) or barrel cortices in mice, and even in smaller animals like flies, whose ommatidia lead to spatially periodic vertical organizations of neurons. So what is special about visual cortex versus these other regions? The lack of discussion of how the present paper relates to other cortical modular structures is unfortunate.

The type of pattern formation we are trying to investigate are neural activity patterns, not anatomical organization. While the laminar structure of cortical columns is present early in development, the degree of the specificity of their lateral organization is poorly understood, although horizontal connections at this age are short range and relatively non-specific as compared to adult horizontal architecture. While mouse barrel cortex has a columnar organization, the functional activity within that network is not modular as we have defined it (distributed, periodic coactivation across distances). Barrel cortex largely reflects the structure of its inputs, with individual whiskers projecting to 'barrelets' in the brain stem, then to 'barreloids' in the thalamus, before arriving in the somatosensory cortex to form 'barrels'. There is little evidence suggesting that these patterns might arise through a dynamic network mechanism utilizing LE/LI, and thus they are

not a good parallel for the modular activity seen in V1.

In the discussion of our manuscript, we do propose that the mechanisms that organize cortical activity in V1 need not be exclusive to visual areas. Any circuit that has local facilitation and effective long-range suppression would be predicted to display modular patterns of neural activity.

Brain development is really a subbranch of biological development and there are lots of examples of cellular patterns forming outside the brain (stripes and spots on the skin, fingerprints, the segmented structure of a spinal cord, formation of five fingers on a hand, etc) and some of these mechanisms are known to be related to Turing instabilities and some involve non-Turing mechanisms like cells that detect absolute concentrations of some morphogen forming a spatial gradient and then the cell undergoes gene activation and specialization when it detects a concentration above or below some threshold (a so-called French-flag model of embryonic patterning).

So the really interesting question the authors should address and discuss is to what extent there are pattern formation mechanisms occurring in brains that do not occur outside the brain. It is remarkable that the authors do not cite any references about biological pattern formation since the Turing instability is just one of many known pattern formation mechanisms.

The reviewer is correct to note that the field of biological pattern formation is quite large. However, a full review of this literature is beyond the scope of our manuscript, and has already been accomplished quite well in recent years (e.g. [17] (which we cite in our manuscript)).

Rather, our manuscript builds upon the large literature (cited in our introduction) that proposes LE/LI interactions underlie modular activity in the visual cortex, a hypotheses we set out to explicitly test. The revisions we have made to our manuscript to re-frame our work directly in terms of these LE/LI mechanisms, and how these mechanisms relate to Turing instability and pattern formation in general, now provide important context for our work. We also now better describe alternative pattern formation hypotheses that could potentially operate in the cortex (see below). Finally, we note that we do both highlight and cite key reviews of biological pattern formation and Turing mechanisms (including [17, 2, 4]).

2. The authors mention way too briefly that “scaffolding” is an alternative theoretical exploration to a Turing instability for explaining modular activity patterns. In a revised paper, the authors should not assume that the readers know what scaffolding is and should add a few sentences of explanation and context.

We have edited the text to provided further detail on ‘scaffolding’ as an alternative hypothesis.

[sic] influenced by lateral boundary conditions that define the boundary of the domain, e.g, boundary conditions can suppress patterns, favor a unique pattern, cause time dependent patterns, etc. The authors completely ignore any consequences of the finiteness of the cortex and any possible influence of boundaries (say caused by changes in the properties of the neurons as one goes from one part of cortex to another). Experiments at some point will be needed to determine if boundaries are having any influence.

The reviewer raises an interesting point, and is correct to note that in certain situations boundary conditions can have a large impact on the behavior of the system. However, it is not clear that such conditions would play a major role in our experiments, or the degree to which such boundaries may exist in the cortex during early development. Our imaging area (maximum 3 mm) is much smaller than the size of ferret visual cortex (over 10 mm in the medial-lateral extent, and approximately an equivalent distance along the anterior-posterior axis after accounting for the folding of cortex around the caudal pole). Thus, any potential across-area boundaries would be some distance beyond our stimulation area, regardless of whether any such boundaries would be accompanied by changes in cellular properties.

That being said, the reviewer is correct to note that the cortex is indeed finite, meaning that boundaries do exist at some point, and future experiments—potentially manipulating activity over much larger regions of cortex—will be required to assess their impacts.

3. The discussion of noise in the paper is too brief and unsatisfying. It would be useful to state briefly what are some sources of noise and try to estimate the magnitudes of these sources. Most pattern formation discussions assume that noise is negligible once a pattern has become nonlinearly saturated but this may not be true for the cortex, especially if synaptic plasticity causes neural interconnections to keep changing in strength.

We have added to the results and discussion examples about potential noise sources within cortical networks that could affect the network state during stimulation. In addition, we note that we do attempt to estimate the magnitude of this noise through the comparison of models with varying noise levels that was presented in Fig S4 (now Fig S5 in revised submission).

4. Figure 2 (and, to some extent, the other figures) are too complex, with too many panels and with too many details included that are never discussed or discussed so briefly that only a few experts know what they mean. (E.g., the meaning of the colors in Fig. 1a, the green and purple wedges in Fig 1b are not mentioned, Fig. 2d is not discussed well in the caption or text and makes little sense, the dice in Fig 1 aren't explain in the caption and not mentioned until long after Figure 1 appears so are just confusing, so just drop these details or figures). I would strongly recommend the authors to reduce the number of panels and details per panel as much as possible. It is the quality of the insight and of the logic that will make this paper interesting and significant, not trying to include every measured or calculated detail, which overwhelms most readers. Any detail shown in a figure should be explained in enough detail to justify to the reader why the detail was included, otherwise, don't include the detail or panel.

We recognize the reviewer's call for simplicity in figure design. We also recognize the need to balance this against the need to show sufficient data and controls to adequately support the claims made in the paper. In general, we favor the inclusion of information and data whenever possible, in order to provide the reader the fullest possible picture of our results.

That being said, we acknowledge that our descriptions of Figures 1 and 2 in our original submission were lacking in some respects and therefore have revised the text and figures to address the specific concerns raised by the reviewer on these figures. We have edited the figure descriptions and removed unnecessary details to make it easier for readers to follow our results. The colors used in Figure 1a and the dice icon used throughout are now labeled within the legend. The colors in Figure 1b were removed, as they were meant to show arbitrary positive/negative growth. The legend for Figure 2d has been expanded and clarified.

Reviewer 2

Mulholland and colleagues conducted simultaneous calcium imaging and optogenetics in the developing ferret visual cortex to test a theoretical model of cortical circuit development. They characterized response of the visual cortex to various spatial patterns of optogenetic stimulation and showed that the experimental results matches with simulations based on the Turing mechanism. They further manipulated activity of the Thalamus and the cortex to show that the optogenetic response patterns were mainly created within the cortex.

While many studies use the mouse as a model system to study cortical circuit development, the mouse brain lacks some important features such as functional columns that are widely observed in other mammalian species including humans. In this study, authors nicely combined cutting edge experimental techniques with sophisticated computational models in the ferret visual cortex and characterized early developmental process of columnar functional circuits. Overall, I think this study is a very important step toward understanding the development of columnar cortical circuits.

Below are my comments.

Major comment.

It was not clear to me how the spatial patterns of optogenetic stimulation affected "self-organization" of cortical activity.

(A) Do spatially structured optogenetic stimulation produce novel spatial patterns not observed in spontaneous activity? Or, (B) do spatially structured optogenetic stimulations just increase occurrence of subsets of spontaneous activity patterns? (similarly to uniform stimulation which increases the occurrence of all spontaneous patterns)

If the latter case (B) were true, is it consistent with the Turing model?

I think the authors can check this point, both for animal experiments and computational simulations, by comparing spatial patterns of *individual* optogenetic events (in Figures 3& 4) with spatial patterns of spontaneous activity (or uniform stimulation).

If the case (B) were true and not consistent with the current Turing model, the authors should weaken the statement that cortical activity patterns are self-organized, throughout the manuscript.

We thank the reviewer for the thoughtful comment, which touches upon a key question in systems neuroscience: What is the nature of the transform between input activity and the output of a cortical network? The two scenarios proposed present two different extremes of the potential mechanisms that could be shaping output patterns. In Scenario A, input patterns are able to shape cortical output and produce novel activity patterns. In Scenario B, the cortical network is rigidly constrained to produce only a specific repertoire of patterns (e.g. those found in spontaneous activity), and the shape of the input has no effect on the output activity except to activate a specific subset of these endogenous patterns. The reviewer is correct to suggest that if the cortex were in such a regime, it would argue against activity patterns within the network being self-organizing.

Following the reviewer's suggestion, we now compare individual spatially structured opto-evoked patterns to the spontaneous patterns we observe in that animal. To assess the overlap between each opto-evoked pattern and this set of spontaneous patterns, we found the spontaneous pattern that was maximally correlated with each opto-evoked pattern and then compared these correlation values to a cross-validated subset of spontaneous vs spontaneous matches. If patterned stimulation only increased the occurrence of subsets of spontaneous activity patterns (Scenario B), we would expect to see no difference in maximum correlations between opto-evoked and control data. Instead, if patterned stimulation generated novel patterns (Scenario A), we would expect to see reduced similarity between the opto-evoked patterns and the repertoire of observed spontaneous events, when compared to controls.

Our analysis, now shown in our revised Figure 6 and supplemental figure 11, shows that the average correlation for best matching opto-spontaneous pairs is weaker than the control spontaneous vs spontaneous correlations (Fig. S11a-c). Additionally, when we project opto-evoked events onto the principal components of spontaneous activity, we found that spontaneous PCs explain less of the total variance of opto-evoked activity compared to spontaneous controls (Fig 6g-h). Together, this indicates that there are novel components in the evoked patterns, supporting Scenario A above. These results are consistent with model predictions (Fig. S11 d-e). Interestingly, our results show that opto-evoked activity patterns are neither perfectly aligned with their stimulus inputs (Figs. 6f, S11a) nor perfectly aligned with endogenous patterns. This indicates that the most extreme version of Scenario A in which inputs entirely dominate cortical activity and fully determine the output pattern—which would also be inconsistent with self-organization within the cortex—is not congruent with our results. Instead, our findings suggest a dynamic compromise between input and the network's inherent tendency to organize activity towards a low-dimensional set of modular patterns. The presence of novel components in patterned opto-evoked activity supports the idea that cortical network shows self-organization of neural activity.

As noted above in our response to reviewer 1, we have also revised our manuscript to be more explicit about what we mean by 'self-organizing'. We show that cortical patterns can arise through local intracortical interactions—as supported by our pharmacological experiments—and can transform optogenetic inputs—as

demonstrated in spatial wavelength transformation experiments of Figure 4 and the input/output pattern transformation of Figures 3 and 6. Collectively, these results support our description of self-organizing neural activity in the cortex.

Minor comments

1) When I first read the manuscript, I did not notice that the optogenetically evoked activity patterns are highpass filtered and mean subtracted. To be able to assess relative strength of filtered and non-filtered activity patterns, please add non-filtered & non-mean subtracted examples, such as an example shown in Fig2d, in Fig3 and Fig4.

We thank the reviewer for this suggestion, and we have included non-filtered and non-mean subtracted examples in supplementary figure S6. These figures show that the modular structure of the neural activity patterns evoked by spatially structured optogenetic stimulation (in Figs 3-4) is clearly evident in the raw $\Delta F/F$ data and does not require any sort of spatial filtering. This is also consistent with the responses evoked from uniform optogenetic stimulation shown in the main text Fig. 2d and in the supplemental movie M1 (showing the non-filtered version of the 5 example trials in Fig. 2e), which is reported as $\Delta F/F$ and is not spatially filtered or mean subtracted.

It is important to note that also neuropil contributes to the calcium signal, and therefore not all signal in between the active domains corresponds to active neurons. Though our opsin was targeted to cell somas, our GCaMP sensor was not, and opto-stimulation seemed to drive global but unstructured increases in fluorescence likely due to neuropil activation. Mean subtracting allowed us to subtract off this constant DC component, while maintaining the trial-varying evoked patterns and allowing us to make more direct comparisons to spontaneous activity patterns. Additionally, in previous work [30] using cellular (two-photon) imaging of early spontaneous activity, we observed little activity in between the active domains during spontaneous events, which suggests that most of what is eliminated through spatial filtering is the neuropil signal.

2) lines887-899 Authors stated "For simplicity, we use a linear rate network model with strong recurrent connections close to the critical value ... the maximum eigenvalue of the connectivity matrix is equal to 0.99". I am curious how sensitive the overall simulation results are to the specific choice of the eigenvalue. Also, it would be nice if the authors could discuss biological plausibility of the developing cortex being at the critical state.

The major goal of the model was to illustrate the effect of driving this class of models with different simulated inputs in the presence of noise. In choosing the parameters for our model, we did a brief assessment of varying the normalization maximum of the eigenvalue of the connectivity matrix (between 0.95 and 0.99) and found that the results of our models are not strongly dependent on having a maximum eigenvalue being close to the critical value. The value of 0.99 was chosen because, given the size, heterogeneity, and amplitude of the input noise, when driven with white noise input to simulate spontaneous (unstimulated) activity, it produced an array of output patterns with a dimensionality similar to that observed in vivo, both from our data and from previously published data [30]. However, networks further away from the critical point tended to produce outputs that are more noisy, with less regularly spaced modules, and in this regard were more similar to those observed in vivo, not inconsistent with the possibility that the cortex may operate some distance away from the critical point. However, from these analyses alone we cannot make firm conclusions about the network state. Our empirical results do not appear to provide a strong constraint specifically on how close to the critical state the cortex operates. While a thorough investigation into how close the network needs to be to the critical state, especially how this relates to other network parameters such as heterogeneity and noise, is an interesting and worthwhile endeavor, it is outside the scope of this particular manuscript.

3) In Fig3f, the values of similarity between individual events and optogenetics spatial patterns seem quite low, hence it is not clear to what extent individual events reflect optogenetics spatial patterns. I think this is partly because only a small fraction of pixels are activated/deactivated in each event (in highpass filtered images). Can the authors calculate correlations using only those pixels that showed large activity change in

each events (e.g. use pixels with $|z| > 2$)?

The reviewer points out an interesting feature of our data: the apparently low amount of similarity between optogenetic patterns that we stimulate with and the pattern that was directly evoked by our stimulus. Although the FOV used for analysis in our experiments are already set conservatively and only include pixels that show both strong visually evoked (through visual change in luminance stimulus) and opto-evoked (from uniform opto-stimuli) responses, it is possible that the relatively low correlation values could be due to including pixels in the analysis that systematically do not show large changes in activity.

Following the reviewer’s suggestion, when we reran our analysis by calculating the correlation for only pixels with large activity changes (pixels with $|z| > 2$). We found that there was a slight increase in correlation value but otherwise our results were not dramatically changed, indicating that the spatial structures of individual events are similar to but not perfectly overlapping with the stimulus pattern.

This result is consistent with what can be observed by looking at the spatial patterns of individual trials in Fig 3b, where there appears to be some overlap with the pattern, but not every module is perfectly aligned and the modules that are activated can change from trial to trial. It is also in agreement with the new analysis in our revised Figure 6, showing that the evoked pattern can contain novel features.

Figure 2: For only pixels with large changes in activity ($|z| > 2$), average similarity of individual opto-evoked trials to their respective stimulus patterns, compared to trial shuffled similarity.

What then could be the cause of this apparent transformation in input pattern to output pattern? One potential explanation is simply that our ability to drive activity is weak with respect to varying sources of noise within the cortex. Within the computational model we also see lower correlations in the similarity between input and stimulus-evoked responses when the relative strength of the stimulus input is weak compared to the noise of the network (supplemental Fig 5h). It is reasonable to think that stimulation of these developing cortical networks might be weak or noisy for a number of reasons: viral expression of the opsin within only a subset of the total population, noisy or unrefined synaptic connectivity within these developing circuits, competing inputs from other feedforward or feedback areas of the brain. All these factors could contribute to evoking responses that deviate somewhat from the stimulus input on an individual trial basis, but whose average responses strongly align to the input pattern (supplemental Fig 5e).

An additional explanation could also potentially be found in the connectivity of the recurrent cortical network. Within random recurrent networks, stimulus inputs that activate preferentially connected subnetworks can amplify responses, which could drive the network to transform activity towards a preferred set of patterns. The stimulation patterns that we are using are artificial and may overlap poorly with the structure of endogenous connectivity within the network. Some heterogeneity in recurrent connectivity can be incorporated in computational models based on LE/LI connectivity, which as a result produce activity patterns with low dimensionality and long-range correlations similar to those seen *in vivo* [30]. Thus, it is possible that the deviation we observe in individual trials from their stimulus input pattern is a dynamic compromise between a LE/LI amplification of activity at a characteristic wavelength and the network’s intrinsic tendency

to drive activity towards preferred patterns via selective amplification of recurrently connected subnetworks. In such a case, one might expect that resulting activity patterns to sit somewhere between stimulus inputs and endogenous activity patterns, consistent with the results we show in the revised Fig 6f.

However, our current data cannot distinguish whether the relatively low degree of similarity we see between stimulus input pattern and individual trial evoked responses is due to either relatively weak input drive or if the specific connectivity of recurrent networks is transforming activity towards preferred responses. Testing this would require many different patterned stimuli that systematically vary the spatial phase of the pattern and quantifying the relationship of those stimuli to the repertoire of spontaneous events. While interesting, such a high resolution mapping of the cortical transform is beyond the scope of the current manuscript.

In our revised manuscript, we now include a larger discussion of the similarity between the input pattern and the resulting activity (Fig. 6f-h), and potential explanations. Additionally, given that calculating the similarity between individual evoked responses and stimulus input patterns using only the most active pixels ($|z| > 2$) did not dramatically change our results, we decided to keep our previous approach and use correlations calculated from all pixels within the field of view, for simplicity of methods purposes.

4) It is not mandatory but it would be nice if the authors could present a data showing that GCaMP and ChrimsonR are co-expressed in single neurons (which I think is the assumption by the authors).

We thank the reviewer for the suggestion, and have now included supplemental figure S2, showing the histology of GCaMP and ChrimsonR expression in developing ferret visual cortex. As would be expected when injecting two separate AAVs, expression of GCaMP and ChrimsonR tended to be intermingled, with some cells expressing both constructs, while others were infected by only one (or none) viruses. This is similar to expression patterns that have been published in other species when using a two virus strategy [7].

One advantage of our single-photon, wide field-of-view approach is that we are targeting and measuring large-scale network effects and do not require single cell resolution. Prior studies [30, 31] found a strong correspondence between the activity of local populations of neurons imaged at the cellular level and the widefield population activity at that location, arguing that widefield imaging captures the average activity of local populations. This prior work also found high levels of coherence in activity amongst local populations of neurons, meaning that even without complete co-expression of sensor and opsin, we can still interrogate how the cortical network responds to relatively large (on the order of hundreds of microns) activation of specific network elements.

5) Y axis in Fig5g and Fig5h should be the same.

We have modified Figure 5 so that the y-axis of Figure 5h matches its control in 5g for ease of comparison.

6) In Fig5i, median of the control is somewhere around 0.2. But in Fig5j, there's no control experiment around Modularity 0.2. Please check.

We thank the reviewer for noticing this discrepancy. Upon review, an error in the file path led to loading data from the wrong experiment trial in Figure 5i, and the figure has now been corrected.

Reviewer 3

This is an impressive manuscript from Mulholland, Kaschube and Smith, which combines widefield fluorescent calcium imaging with optogenetic stimulation to show that modular responses emerge from cortical interactions in the developing ferret visual cortex (V1). This work is motivated by the prior finding that modular activity patterns at the spatial scale of cortical columns are present in the spontaneous activity of visual cortex prior to eye opening. They propose a neural implementation which requires recurrent excitation of neighboring neurons -which form a module- combined with longer range inhibition between modules.

Such a model generates periodic groups of co-activated modules which resemble the periodic functional maps observed in primate and carnivore neocortex. The authors describe 3 predictions of the Turing model, each of which they test by optogenetically stimulating ferret V1 prior to eye opening. First, they show that non-specific activation of V1 with optostim produces modular and periodic patterns of neural activation, in line with the Turing prediction. Next, they stimulated V1 with random patterns at the characteristic wavelength of cortical modules observed during spontaneous activity. Indeed the trial-averaged activation patterns in V1 strongly resembled the spatial structure of the optostim. To further test the validity of this candidate mechanism, they presented optostim patterns at spatial wavelengths which differed from the characteristic wavelength of cortical modules observed during spontaneous activity. Their model predicts that inputs away from the characteristic wavelength will produce a response at an intermediate wavelength. They show that indeed, the spatial wavelength of cortical activation following patterned stimulation lies at a value between the input and characteristic wavelengths, and that this effect is present when input wavelength is both narrower and broader than the characteristic wavelength. This final point is quite valuable, as it shows that modular patterns of cortical activation do not simply reflect the spatial pattern of inputs, nor are they confined to a single wavelength regardless of input.

An alternative hypothesis for modular responses is that the structure of the feedforward activity is essential. The authors reject this hypothesis by showing that modular activation following non-specific optostim persists when LGN is silenced with Muscimol.

Finally, the authors compare the optostim-evoked activation patterns to the spontaneous patterns which they had previously observed in cortex. They find that these patterns are highly correlated in their spatial arrangement and can be accounted for by the same low-dimensional principal component representation. This result indicates that these patterns emerge from a common synaptic source- an important finding in support of their hypothesis that a Turing-like organization seeds the formation of modular, periodic maps in cortex.

This is a very strong manuscript. The authors lay out a clear hypothesis – that the Turing-like mechanism describes the population-level organization of ferret cortex prior to eye opening. They make testable predictions of how a neural population organized in this way would respond to different patterns of input. They show that each of these predictions is met in-vivo. This work represents a significant advance in our understanding of cortical organization early in development. It also constrains models of functional maps in cortex by indicating that columnar organization is an emergent property of the structure of intracortical connections present prior to visual experience, rather than depending on selective pooling of peripheral afferents.

I have only one critical comment for this work. The manuscript seeks to describe the neural mechanisms which generates modular organization in the absence of structured visual input. And yet there may be many circuit mechanisms that implement this computational model. Critically some of the circuit models that the authors cite, for example, the attractor models of Sompolinsky, exhibit dynamics that do not seem to match cortical networks. Are there any aspects of the cortical dynamics from the present recordings that could make constraints on the neural implementation of this Turing-like phenomenon?

We thank the reviewer for their comments and their enthusiasm for our work. The attractor models by Sompolinsky and others of the ring-model type are related to the model class used in this study. However, our model does not operate in the so-called ‘marginal state’ (corresponding to a supercritical regime, [5]), but in the ‘linear’ state (subcritical), however close to the boundary of linear instability. In either of these regimes pattern formation involves cross-correlations over relatively long time scales (on the order of one hundred milliseconds), which may be unrealistic in the mature cortex [16], where balanced amplification [23] seems to be a plausible alternative. Although we note, these attractor-type models may be less problematic in the early cortex, where circuits are built and the inputs from the LGN are still temporally imprecise, [1]. Unfortunately, however, the calcium sensor we are using, selected for its high sensitivity and good signal-to-noise, does not provide sufficient time resolution to distinguish between these different possible models. It would be interesting to repeat our experiments in the future with a faster readout of activity, for instance using neuropixels or voltage-sensitive imaging.

While our experiments provide strong evidence for an intracortical interaction that is effectively LE/LI, drawing further conclusions on the specific network constraints on the neural implementation of this mechanism is challenging. Due to our widefield 1-photon population level imaging, our data provide little insights about the anatomical connections implementing this model. Future pharmacology experiments or targeted manipulations of individual circuit components using cellular resolution optogenetics could have the potential to reveal the importance of different circuit elements. It would also be interesting to repeat our experiments at different stages in development (and with a faster readout) to assess whether the effective intracortical interactions change during cortical maturation, for instance transitioning from a more recurrent to a more input-dominated regime.

With our existing data, however, our experiments do provide new empirical evidence that may constrain future models. Previous experiments conducted at a similar age range in ferret visual cortex showed that spontaneous excitatory activity became abnormally large when suppressing inhibition [22], which indicates that the cortex, at this age, operates in a regime sometimes called a inhibition-stabilized network (ISN) [27]. This is also the regime of the model in this manuscript. For certain conditions, ISNs show a dependency of the spatial wavelength as a function of stimulus drive [26]. Our results show that the spatial wavelength is virtually unchanged as a function of laser power (which is also what the model used in this study would predict), providing an important constraint that was previously not emphasized in our manuscript.

We thank the reviewer for their suggestion, and now include an expanded discussion of attractor network models in our revised discussion.

References

- [1] Colin J Akerman, Matthew S Grubb, and Ian D Thompson. Spatial and temporal properties of visual responses in the thalamus of the developing ferret. *Journal of Neuroscience*, 24(1):170–182, 2004.
- [2] Anika Anirban. 70 years of turing patterns. *Nature Reviews Physics*, 4:432, 2021.
- [3] Tanya I Baker and Jack D Cowan. Spontaneous pattern formation and pinning in the primary visual cortex. *Journal of Physiology-Paris*, 103(1-2):52–68, 2009.
- [4] Phillip Ball. The self-made tapestry: pattern formation in nature. 1999.
- [5] R Ben-Yishai, R L Bar-Or, and Haim Sompolinsky. Theory of orientation tuning in visual cortex. *Proceedings of the National Academy of Sciences*, 92(9):3844–3848, 1995.
- [6] William H Bosking, Ying Zhang, Brett Schofield, and David Fitzpatrick. Orientation selectivity and the arrangement of horizontal connections in tree shrew striate cortex. *Journal of Neuroscience*, 17(6):2112–2127, 1997.
- [7] Selmaan N Chettih and Christopher D Harvey. Single-neuron perturbations reveal feature-specific competition in v1. *Nature*, 567(7748):334–340, 2019.
- [8] P G Clarke, I M Donaldson, and D Whitteridge. Binocular visual mechanisms in cortical areas i and ii of the sheep. *Journal of Physiology*, 256(3):509–526, 1976.
- [9] Jeremy C Durack and Lawrence C Katz. Development of horizontal projections in layer 2/3 of ferret visual cortex. *Cerebral Cortex*, 6(2):178–183, 1996.
- [10] Joshua A Goldberg, Uri Rokni, and Haim Sompolinsky. Patterns of ongoing activity and the functional architecture of the primary visual cortex. *Neuron*, 42(3):489–500, 2004.
- [11] Agnieszka Grabska-Barwińska and Christoph von der Malsburg. Establishment of a scaffold for orientation maps in primary visual cortex of higher mammals. *Journal of Neuroscience*, 28(1):249–257, 2008.

- [12] Antony M Grigonis, Gloria J Zingaro, and E. Hazel Murphy. The development of orientation and direction selectivity in the rabbit visual cortex. *Developmental Brain Research*, 40(2):315–318, 1988.
- [13] David H Hubel and Torsten N Wiesel. Functional architecture of macaque monkey visual cortex. *Proceedings of the Royal Society of London. Series B. Biological Sciences*, 198(1130):1–59, 1977.
- [14] Michael Ibbotson and Young Jun Jung. Origins of functional organization in the visual cortex. *Frontiers in Systems Neuroscience*, 14(10):1—13, 2020.
- [15] Matthias Kaschube, Michael Schnabel, Siegrid Löwel, David M Coppola, Leonard W White, and Fred Wolf. Universality in the evolution of orientation columns in the visual cortex. *Science*, 330(6007):1113–1116, 2010.
- [16] Mikail Khona and Ila R Fiete. Attractor and integrator networks in the brain. *Nature Reviews Neuroscience*, 23(12):744–766, 2022.
- [17] Shigeru Kondo and Takashi Miura. Reaction-diffusion model as a framework for understanding biological pattern formation. *Science*, 329(5999):1616–1620, 2010.
- [18] Alexei A Koulakov and Dmitri B Chklovskii. Orientation preference patterns in mammalian visual cortex: A wire length minimization approach. *Neuron*, 29:519—527, 2001.
- [19] Guang Bin Liu, Donaldson, and John D Pettigrew. Orientation mosaic in barn owl’s visual wulst revealed by optical imaging: comparison with cat and monkey striate and extra-striate areas. *Brain Research*, 961(1):153–158, 2003.
- [20] Hans Meinhardt and Alfred Gierer. Applications of a theory of biological pattern formation based on lateral inhibition. *Journal of cell science*, 15(2):321–346, 1974.
- [21] Hans Meinhardt and Alfred Gierer. Pattern formation by local self-activation and lateral inhibition. *Bioessays*, 22(8):753–760, 2000.
- [22] Haleigh N Mulholland, Bettina Hein, Matthias Kaschube, and Gordon B Smith. Tightly coupled inhibitory and excitatory functional networks in the developing primary visual cortex. *eLife*, 10:1–17, 2021.
- [23] Brendan K Murphy and Kenneth D Miller. Balanced amplification: A new mechanism of selective amplification of neural activity patterns. *Neuron*, 61(4):635–648, 2009.
- [24] Benedict Shien Wei Ng, Agnieszka Grabska-Barwińska, Onur Güntürkün, and Dirk Jancke. Dominant vertical orientation processing without clustered maps: early visual brain dynamics imaged with voltage-sensitive dye in the pigeon visual wulst. *Journal of Neuroscience*, 30(19):6713–6725, 2010.
- [25] Kenichi Ohki, Sooyoung Chung, Yeang H Ch’ng, Prakash Kara, and R Clay Clay. Functional imaging with cellular resolution reveals precise micro- architecture in visual cortex. *Nature*, 433:597–603, 2005.
- [26] Daniel B Rubin, Stephen D Van Hooser, and Kenneth D Miller. The stabilized supralinear network: A unifying circuit motif underlying multi-input integration in sensory cortex. *Neuron*, 85(2):402–417, 2015.
- [27] Sadra Sadeh and Claudia Clopath. Patterned perturbation of inhibition can reveal the dynamical structure of neural processing. *eLife*, 9:1–29, 2020.
- [28] Kerstin E Schmidt and Fred Wolf. Punctuated evolution of visual cortical circuits? evidence from the large rodent *dasyprocta leporina*, and the tiny primate *microcebus murinus*. *Current Opinion in Neurobiology*, 71:110–118, 2021.
- [29] Harel Z Shouval, David H Goldberg, Judson P Jones, Martin Beckerman, and Leon N Cooper. Structured long-range connections can provide a scaffold for orientation maps. *Journal of Neuroscience*, 20(3):1119–1128, 2000.

- [30] Gordon B Smith, Bettina Hein, David E Whitney, David Fitzpatrick, and Matthias Kaschube. Distributed network interactions and their emergence in developing neocortex. *Nature neuroscience*, 21(11):1600–1608, 2018.
- [31] Gordon B Smith, David E Whitney, and David Fitzpatrick. Modular representation of luminance polarity in the superficial layers of primary visual cortex. *Neuron*, 88(4):805–818, 2015.
- [32] Nicholas V Swindale. A model for the formation of orientation columns. *Proceedings of the Royal Society of London - Biological Sciences*, 215(1199):211–230, 1982.
- [33] Nicholas V Swindale. Coverage and the design of striate cortex. *Biological Cybernetics*, 65(6):415–424, 1991.
- [34] Stephen D Van Hooser, J. Alexander F Heimerl, Sooyoung Chung, Sacha B Nelson, and Louis J Toth. Orientation selectivity without orientation maps in visual cortex of a highly visual mammal. *Journal of Neuroscience*, 25(1):19–28, 2005.
- [35] Christoph von der Malsburg. Self-organization of orientation sensitive cells in the striate cortex. *Biological Cybernetics*, 14:85–100, 1973.
- [36] Marvin Weigand, Fabio Sartori, and Hermann Cuntz. Universal transition from unstructured to structured neural maps. *Proceedings of the National Academy of Sciences*, 114(20):4057–4064, 2017.
- [37] Lewis Wolpert, Cheryll Tickle, and Alfonso Martinez Arias. *Principles of development*. Oxford University Press, USA, 2015.

REVIEWERS' COMMENTS:

Reviewer #1 (Remarks to the Author):

See attachment.

**Nature Communications Referee Report for Manuscript
444555_1**

**“Self-organization of modular activity in immature cortical
networks”**

by H. Mulholland et al

January 23, 2024

I find this revised manuscript acceptable for publication without further changes.

I read carefully the reply to the editor by the authors and the revised manuscript but only quickly scanned through the supplementary material. I found that the authors were careful and thorough in thinking about and then doing a very good job in responding to comments of the three referees. In particular, I feel that the authors did a very good job in addressing my comments (as Reviewer 1) and I appreciate and am happy with their changes. I especially think that the paper was improved by emphasizing the central role of LE/LI connections rather than Turing patterns. As a result of the many changes, I feel the paper is stronger and easier to understand and appreciate.

I would like to bring to the attention of the authors three optional minor changes that the authors could consider making before this paper is published, all dealing with making certain clarifications rather addressing the experimental and theoretical results:

1. The way the paper is written, with the phrase LE/LI repeated so many times, the reader may think incorrectly that LE/LI connectivity is an established well-understood fact in visual cortex. But as the authors point out near line 25 on page 15 of their revised manuscript, currently there is only indirect anatomical evidence of LE/LI, and details are currently lacking regarding which neurons are involved, what is the connectivity of these neurons, etc. I think this point should be stated more clearly, in the introduction and conclusion, since readers might miss this point.
2. As currently written and in several places, the paper is a bit too strong in claiming that they have demonstrated that immature cortical networks have self-organized activity when, in fact, what the paper has tested are three indirect aspects of self-organization based on a highly simplified and idealized linear model as summarized in Figure 1. For example, around line 5 on page 16 of the revised paper, the authors say “... our work demonstrates the power of immature cortical networks to self-organize neural activity, ...” but I would recommend restating this as saying that the results of this paper make the hypothesis of self-organization of neural activity based on an LE/LI mechanism much more plausible, with the implication that this paper should stimulate further near-term research to collect enough new details to settle this.

I do not feel that this paper is a definitive demonstration of LE/LI leading to self-organization (although I think it is the most likely possibility and I cannot think of simple alternatives). A more definitive future paper would likely involve using some combination of dense connectomics of the V1 region (or maybe Ed Callaway-like sparse connectomics) combined with more detailed nonlinear biophysical simulations of the dense connectomics to show that specific experimental circuitry, when simulated, leads to the experimentally observed self-organization of neural activity.

3. I think it would help if the paper added a few sentences at the end of the discussion on page 16, saying that a major remaining mystery is why some animals do not have modular activity, they have a salt-and-pepper mixed architecture. That is, while this paper makes a strong case that LE/LI leads to modular cortical activity for young ferrets and so presumably for similar animals, the paper doesn't discuss that the significance of this modular activity — why these animals evolved to have this capability but other visual animals did not — is not understood. LE/LI could perhaps be a consequence of brain-size-related optimizations not related to behavior such as minimizing wire length and reducing delay times, which the authors mention briefly around line 35 on page 14, but this point is buried towards the end of a long paragraph and instead should be more prominent.

This is the kind of point that might be settled by future theoretical evolutionary and optimization calculations of the kind that James DiCarlo and collaborators (and others) have carried out, by studying layers of artificial neural networks with few imposed details, and by imposing some constraints roughly consistent with experiment (such as maybe a 4:1 ratio of excitatory to inhibitory neurons, certain probabilities of connectivity as a function of radius, etc), and then seeing what kind of dynamics evolves for certain classes of stimuli. The linear pattern formation model used by the authors is, while insightful, phenomenological and does not clarify why some animals did and did not evolve to have modular activity.

Again, I feel that making changes in response to these comments is optional, the paper is acceptable in its current form.

Reviewer #2 (Remarks to the Author):

The authors have addressed all my concerns. They have notably performed further analysis of opto-evoked activity on a single-trial basis. Additionally, they acknowledged that their data does not entirely support the interpretation of the developing cortex being in a critical state. I have no additional comments. This paper makes a substantial contribution to the field.

Reviewer #3 (Remarks to the Author):

The authors have addressed my concerns.

We thank all 3 reviewers for their careful consideration of our manuscript. All reviewers now find the manuscript acceptable for publication, without further changes. Reviewer 1 raises some additional points for optional further edits, which we agree would further strengthen the manuscript. Below we address the comments of all reviewers in detail.

Reviewer 1

I find this revised manuscript acceptable for publication without further changes.

I read carefully the reply to the editor by the authors and the revised manuscript but only quickly scanned through the supplementary material. I found that the authors were careful and thorough in thinking about and then doing a very good job in responding to comments of the three referees. In particular, I feel that the authors did a very good job in addressing my comments (as Reviewer 1) and I appreciate and am happy with their changes. I especially think that the paper was improved by emphasizing the central role of LE/LI connections rather than Turing patterns. As a result of the many changes, I feel the paper is stronger and easier to understand and appreciate. I would like to bring to the attention of the authors three optional minor changes that the authors could consider making before this paper is published, all dealing with making certain clarifications rather addressing the experimental and theoretical results:

We are pleased to hear that the changes we made were able to address the concerns raised in our initial submission, and agree that the manuscript is now much stronger for it. We are glad that Reviewer 1 now finds the manuscript acceptable for publication without further changes.

In addition, we also thank the reviewer for raising further 3 points in their comments, and their suggestions for optional further edits to the manuscript. We agree with these points, and have made additional changes to the manuscript as noted below.

1. The way the paper is written, with the phrase LE/LI repeated so many times, the reader may think incorrectly that LE/LI connectivity is an established well understood fact in visual cortex. But as the authors point out near line 25 on page 15 of their revised manuscript, currently there is only indirect anatomical evidence of LE/LI, and details are currently lacking regarding which neurons are involved, what is the connectivity of these neurons, etc. I think this point should be stated more clearly, in the introduction and conclusion, since readers might miss this point.

Reviewer 1 is correct in that LE/LI is mostly a hypothesized connectivity scheme, which is currently lacking in clear anatomical evidence. To make this clearer for the reader, we added a line in the conclusion that more explicitly states this (page 10, line 20-21), and state that LE/LI is a hypothesized mechanism in the introduction. Also, we highlight that paragraph 4 of our introduction also already makes this point ('empirical evidence for specific neural connectivity schemes that could support such a mechanism has been scarce'). We believe these changes clarify this important point for readers.

2. As currently written and in several places, the paper is a bit too strong in claiming that they have demonstrated that immature cortical networks have self-organized activity when, in fact, what the paper has tested are three indirect aspects of self-organization based on a highly simplified and idealized linear model as summarized in Figure 1. For example, around line 5 on page 16 of the revised paper, the authors say ". . . our work demonstrates the power of immature cortical networks to self-organize neural activity, . . ." but I would recommend restating this as saying that the results of this paper make the hypothesis of self-organization of neural activity

based on an LE/LI mechanism much more plausible, with the implication that this paper should stimulate further near-term research to collect enough new details to settle this.

I do not feel that this paper is a definitive demonstration of LE/LI leading to self-organization (although I think it is the most likely possibility and I cannot think of simple alternatives). A more definitive future paper would likely involve using some combination of dense connectomics of the V1 region (or maybe Ed Callaway-like sparse connectomics) combined with more detailed nonlinear biophysical simulations of the dense connectomics to show that specific experimental circuitry, when simulated, leads to the experimentally observed self-organization of neural activity.

We have softened some of the language of our conclusions. Our results are consistent with the predictions made by this class of models, but at this time we cannot entirely rule out alternative mechanisms—although our results are inconsistent with current alternative theories, such as retinal mosaics or a ridged, clustered scaffold. LE/LI seems like the most plausible theory, but much work still needs to be done to more fully understand whether and how this mechanism is implemented in the cortex. In the discussion we acknowledge that a more thorough understanding of the connectomics in the mammalian cortex may reveal new insights into how immature cortical circuits transform inputs.

While we acknowledge that the idealized linear model that we used to do some simulations in this paper is likely oversimplified, it is the simplest demonstration of the key phenomenon that we investigated in this study, which is that outputs of LE/LI networks are biased towards a characteristic wavelength. Importantly, the emergence of modular outputs has been shown in nonlinear, more biologically plausible models e.g. in (Smith et al., 2018, Antolik 2017), indicating that lateral interactions can shape network outputs under a diverse range of network parameters. It would be interesting to explore whether similar results would be expected in nonlinear models guided by connectomics, as this would provide a more detailed set of predictions about which neurons are involved in producing these patterns of activity.

3. I think it would help if the paper added a few sentences at the end of the discussion on page 16, saying that a major remaining mystery is why some animals do not have modular activity, they have a salt-and-pepper mixed architecture. That is, while this paper makes a strong case that LE/LI leads to modular cortical activity for young ferrets and so presumably for similar animals, the paper doesn't discuss that the significance of this modular activity — why these animals evolved to have this capability but other visual animals did not — is not understood. LE/LI could perhaps be a consequence of brain-size-related optimizations not related to behavior such as minimizing wire length and reducing delay times, which the authors mention briefly around line 35 on page 14, but this point is buried towards the end of a long paragraph and instead should be more prominent.

This is the kind of point that might be settled by future theoretical evolutionary and optimization calculations of the kind that James DiCarlo and collaborators (and others) have carried out, by studying layers of artificial neural networks with few imposed details, and by imposing some constraints roughly consistent with experiment (such as maybe a 4:1 ratio of excitatory to inhibitory neurons, certain probabilities of connectivity as a function of radius, etc), and then seeing what kind of dynamics evolves for certain classes of stimuli. The linear pattern formation model used by the authors is, while insightful, phenomenological and does not clarify why some animals did and did not evolve to have modular activity.

We agree that this is an important ongoing question, and we have added a paragraph to the discussion expanding on this topic. Why cortical networks exhibit spatial clustering of activity and neural representations is still poorly understood. Is it merely an epiphenomenon of a wire-

length minimization problem, or does it confer other computational benefits? The proposed work above could shed some interesting light on these topics, especially if it could explain some of the differences found between primates/carnivores and rodents. This would be especially interesting in the context of the growing evidence that rodent cortical networks do exhibit some modest degree of spatial organization, but on much smaller scales, and if these sorts of models could simulate what effects these changes in scale might have on computations.

Again, I feel that making changes in response to these comments is optional, the paper is acceptable in its current form.

Reviewer #2 (Remarks to the Author):

The authors have addressed all my concerns. They have notably performed further analysis of opto-evoked activity on a single-trial basis. Additionally, they acknowledged that their data does not entirely support the interpretation of the developing cortex being in a critical state. I have no additional comments. This paper makes a substantial contribution to the field.

Reviewer #3 (Remarks to the Author):

The authors have addressed my concerns.

We thank reviewers 2 & 3 for their efforts in reviewing our manuscript, and are pleased to hear that our revisions have addressed all of their concerns.